

# Hydrofacies reconstruction of glaciofluvial aquifers and groundwater flow modelling in a densely urbanized area under changing climatic conditions

Mattia De Caro[1], Giovanni B. Crosta[1], Paolo Frattini[1], Roberta Perico[1], Giorgio Volpi[1]

[1]DISAT – Department of Earth and Environmental Sciences, University of Milano Bicocca, Piazza della Scienza, 4, Milan 20126, Italy

*Correspondence to*: Mattia De Caro (m.decaro@campus.unimib.it)

**Abstract.** Potential impacts of global climate changes on the groundwater are largely unknown, especially for densely populated areas where groundwater is heavily exploited for public and industrial supply. Hence, to better plan and manage the groundwater resources, medium-long term numerical modelling of groundwater flow, which takes into account climate change, population growth, and industrial and agricultural activities, is needed. The objective of this paper is to tackle three main issues: (1) the development of a robust hydro-stratigraphic model of a multi-aquifer system from a well logs database by means of a novel multi-dimensional approach which includes a hierarchical classification of the lithologies, the interpretation of cross-sections, and the interpolation of aquifer boundary surfaces; (2) the parametrization and calibration of both a steady state and a transient groundwater flow model, starting from empirical relationships (for unconfined aquifer) and step-drawdown and well tests (for semi-confined and confined aquifers) to define equivalent homogenous sub-units; and (3) the simulation of steady state and transient scenarios based on projections about global climate change and variation in abstraction and recharge rates. These issues are illustrated for the Milan metropolitan area (Northern Italy) and the conterminous Po Plain portion. The results of the model allow to analyse the major components of the regional groundwater system (i.e. public supply wells withdrawals, discharge to gaining rivers and springs, recharge from irrigation networks and vegetated areas, flow transfer between aquifers). The groundwater level rising observed in the last decades caused serious problems in the urban areas and a progressive increase in the base-flow towards the gaining rivers. Simulations including effects of future climate scenarios (2017–2030) indicate a further increase in groundwater level in the next decades at a lower rate (ca. 0.3 m/year) with respect to that of the 1970–2016 period (ca. 1 m/year), due to the combined action of decreasing withdrawals and recharge.

## 1 Introduction

The study of groundwater in densely populated areas is motivated by the strong interaction between the socio-economic development and the groundwater environmental impacts, the urbanization and agricultural activities affecting both the



availability and the quality of the groundwater resources. It has been long recognised that urbanization results in important

changes to the groundwater budget. Abstraction from wells and changes in land use replace and modify groundwater flow by introducing new discharge and recharge patterns (Foster, 2001). Main water supply and sewage systems can have a significant impact on shallow aquifers that underlie a city because of leakage and seepage. Where a city relies on aquifers located within densely populated areas for a significant component of its water supply requirements, the groundwater bodies may be subjected to depletion and a deterioration in quality (Morris et al., 1994). As the water demand of industrial sector has fallen, many cities

are now experiencing rising groundwater levels (e.g. Melbourne, Mudd et al., 2004; Tokyo, Hayashi et al., 2009; Kuwait, Doha, Cairo, Riyadh, Jizzan, Tabrik, Buraidh, Madinah and Jubail, George 1992; Buenos Aires, Hernandez et al., 1997; Barcelona, Vazquez-Sune et al., 2005; Jeddah, Al-Sefry and Şen, 2004; Liverpool, London, Wilkinson, 1985; Lerner and Barrett, 1996; Paris, Lamè, 2013) with consequent concerns about damage to subsurface engineering structures, as a result of hydrostatic uplift or reduced bearing capacity, inundation of subsurface facilities, excessive ingress of groundwater to sewers,

chemical attack on concrete foundations, and the mobilization of contaminants (Foster, 2001; Lelliott et al., 2006, Brassington & Rushton 1987; Brassington 1990; Wilkinson & Brassington 1991; Knipe et al. 1993; Greswell et al. 1994; Heathcote & Crompton 1997; Cheney, 1999).

It is widely recognised that global changes can significantly affect water resources (IPCC, 2008; Green et al., 2011; Fung and Lopez, 2011), but research has been focused mainly on surface water (IPCC, 2008), while little is known about the potential

impact of global change on groundwater (Green et al., 2011; Taylor et al., 2013). Projections from the Intergovernmental Panel on Climate Change (IPCC) show relevant global warming and changes in the precipitation patterns during the 21$^{st}$ century (Rogelj et al., 2012). These changes are expected to affect the hydrological cycle, altering groundwater levels and recharge with various associated impacts on natural ecosystem and human activities (Bates et al., 2008; Green et al., 2011; Taylor et al., 2013). Climate studies agree in prospecting a general decrease of average precipitation, an increase in extreme meteoric

events, and an increase of mean temperature in Northern Italy south of the Alps (Gobiet et al, 2014). In this context, an accurate and realistic hydrogeological model is needed to produce indicative results, taking the aquifer characteristics into account and considering the transient behaviour of the whole hydrological system. The reconstruction of the aquifer geometry is pivotal to the development of such model. For heterogeneous deposits, such as glaciofluvial deposits, this can be achieved by using descriptive, structure-imitating and process-imitating techniques (Kolterman and Gorelick, 1996). At present day, numerical

models could handle detailed data on aquifer heterogeneities; nevertheless defining the heterogeneity is elusive as conventional hydrogeological data are generally insufficient to characterize the actual distribution of hydraulic conductivity (Rubin and Hubbard, 2005). Therefore, an approach employing common available data (i.e. borehole logs, well tests, grain size analyses) to identify and correlate hydrofacies in space, and thus to estimate hydraulic conductivity over different nearly-homogenous units of a 3D numerical groundwater flow model is of great value for practical applications (Ouellon et al., 2007).

In the Milan metropolitan area (Northern Italy), the groundwater has been heavily exploited for public and industrial supply. The rate of abstraction decreased during the last century, with the number of inhabitants, the diminished *per capita* consumption and the decommissioning of industrial activities. This resulted in a water table rise inducing flooding of deep



constructions (e.g. building foundations and basements, subways, excavations, gravel pits). On the other hand, a reduction of groundwater recharge (Facchi et al., 2005) could results from climate changes and that may partially compensate for the decrease of abstraction in the following decades.

Several groundwater models have been developed for the Adda-Ticino area (Giura et al, 1995; Canepa, 2011; Alberti et al., 2016) and for some specific portions (Giudici et al., 2000, 2001; Facchi et al., 2004; Vassena et al., 2012), including recharge estimation and hydraulic conductivity calibration. Steady-state models have been developed (Bonomi, 2009; Canepa, 2011; Vassena et al., 2012; Bonomi et al., 2014; Alberti et al., 2016) considering the aquifer groups definition given by Regione Lombardia and Eni (2002). In addition, some essential hydrological features (i.e. well withdrawals and spring discharge) have been neglected in these models and this may lead to significant discrepancies in the estimated hydrogeological budget and model predictions. In this research, a large-scale groundwater model capable to capture the overall groundwater dynamics (both in steady and transient state) based on a novel aquifers reconstruction and on a comprehensive hydraulic parametrization is developed.

The textural variations of the Adda-Ticino area (or subdomains) at different scales have been studied by hierarchical simulation procedures, based on binary tree approach (Comunian et al., 2016), and by combining geological and geophysical data (Mele et al., 2010, 2012, 2013). In this research, starting from hierarchical classification (i.e. lithofacies, hydrofacies and aquifer groups) of glaciofluvial deposits in front of the ice margin (Miall, 1978; Anderson, 1989; Miall, 1988; Cavalli, 2012), a new hydrostratigraphic model of the Adda-Ticino area is developed by means of a deterministic multi-dimensional approach. Hydrofacies are defined as granular material of similar grain size and hydraulic properties (Ouellon et al., 2008).

This study addresses several topics: (i) hydrostratigraphic modelling for the reconstruction of the aquifers geometry of glaciofluvial aquifers, (ii) parametrization of the aquifers by means of well test analyses and grain-size-distribution/hydraulic conductivity empirical relationships, (iii) processes identification, including sink/sources characterization, (iv) groundwater flow modelling including steady and long term transient-state finite element simulations and, (v) simulation of future scenarios based on projections about global climate change and variation in abstraction and recharge rates.

## 2 Study area

### 2.1 Geographical and geomorphological settings

The study area (Fig. 1) is located in the Po Plain (Northern Italy) and covers a portion of 3 135 km$^2$. The area is bounded by the Po River to the South, the Adda and the Ticino rivers to the East and West, respectively, and to the North by the Prealps foothills. The entire study area is here defined as Milan metropolitan area and consists of urban and rural areas located around Milan, with a strong socio-economical link with the main city. The area hosts about 8.5% of the Italian population (5 181 192 people, 3 208 509 of which in the Metropolitan City of Milan, ISTAT, 2016). From a geomorphological point of view, the study area consists of glacial terraces and alluvial fans placed below the frontal moraines at the outlets of the Alpine valleys. The area can be divided into high and low plain. The first is characterized by coarser deposits and a slope of about 0.005, the



latter, by fine alluvial deposits with a slope of about 0.002. The climate is continental and the mean annual precipitation ranges from 600 mm/year to 800 mm/year in the lower plain. The Adda, the Ticino and the Lambro valleys are deeply incised (5 to 30 m) due to erosion of post-glacial deposits, leading to several orders of fluvial terraces. Consequently, the rivers flow at a lower level than the regional groundwater level resulting prevalently in a gaining river condition (Giudici et al., 2007).

## 2.2 Geological settings

The Po Plain represents the peripheral foreland basin of the Alps, developed during the Pliocene in response to global climate change forcing the sedimentation rate and Alps uplift. Recently, knowledge of the Po Plain superficial deposits has been improved through high resolution seismic (Francese et al., 2005; De Franco et al., 2009), electric borehole logs, and lithostratigraphic and petrographic analyses on 11 deep (200 m) boreholes (Regione Lombardia and ENI, 2002; Scardia et al., 2006, 2010; Garzanti et al., 2011). Three main depositional sequences have been recognised (Garzanti et al., 2011; Scardia et al., 2012) corresponding to as many aquifers, which can be summarized as follow (from bottom to top):

I. The deep confined aquifer consists of sandy lenses within clay and silt units representing the lower Pliocene continental-marine facies. This sequence (PS1, late Early Pleistocene, 1.4–0.87 Ma) mainly consists in meandering river plain deposits fed from the Western and Central Alps (Muttoni et al., 2003; Garzanti et al., 2011), and prograding axially in low subsidence settings. The aquifer base consists of Pliocene continental-marine deposits.

II. The semi-confined aquifer consists of sands and sandy gravels with a thickness in the range between 50 m and 150 m. The lower portion of the aquifer consists of clay and silt layers, and locally of conglomeratic units (locally known as Ceppo). The major Pleistocene glaciations in the Alps produced a major sequence boundary in the Po basin at 0.87 Ma. This surface marks the synchronous and widespread progradation of the braid-plain sequences (PS2) over the previous meandering river deposits. This second sequence corresponds to the distal fringe of the glacial outwash plains which transversally prograde moving southward (Garzanti et al., 2011).

III. The unconfined aquifer consists mainly of gravel with a sandy matrix. The aquifer, with a thickness in the range between 20 m and100 m, overlays a clayey silty aquitard which shows a good continuity in the southern portion of the study area, whereas it disappears moving northward. The inferred age is about 0.45 Ma (Regione Lombardia and Eni, 2002) and corresponds to a regional discontinuity. The sequence (PS3) has been developed during the Middle-Late Pleistocene and consists of proximal braid-plain deposits composed mainly of coarse and poorly sorted gravels.

To the South, in proximity of S. Colombano (Fig. 1), the syndepositional tectonics related to the activity of an anticline ramp above the Apenninic blind thrusts uplifted the Pliocene and the Lower Pleistocene basement (Bersezio et al., 2010).

Recently, the compositional variations of the Po-plain deposits have been related to the *LGM* (Last Glacial Maximum) and post *LGM* evolution of alluvial megafans and fans (Fontana et al., 2014). The development of these fans is mainly related to climatic variations. During the *LGM*, the development of the sequences was controlled by changes in the sediment supply



and water discharge ratio. Therefore, since the middle Pleistocene, the depositional system evolved through sedimentary pulses (glacial maxima) alternated with interglacial periods of stability, leading to regional discontinuities at the transition between phases (Fontana et al., 2014). Accordingly, several fan systems have been distinguished in the study area (Lambro

megafan, Seveso fan, Olona megafan, Lura fan, and Molgora megafan, Fontana, 2014). During the *LGM* the western branch of Como Lake hosted a glacial front that supplied the Lura and Seveso systems. Together with the Lambro, these rivers formed the Milan plain, but they were constrained and partially buried by the Olona megafan, which was the main outwash of the glacier hosted in the Lugano basin (Bini, 1997). After glacial withdrawal the Lura, the Seveso and the Olona became minor streams and their alluvial systems were abandoned.

## 135  2.3 Hydrogeological settings

The study area is characterized by a complex hydrographic network (Fig. 1). The Ticino and the Adda rivers flow from the Maggiore and the Lario lake respectively, to the Po river. These rivers are considered in contact with the shallow water table (i.e. gaining rivers). In fact, after glacial withdrawal, the Adda and the Ticino rivers formed incised valleys (5 m to 30 m deep). The Olona, the Seveso and the Lambro rivers flow in the area. Among these, the Lambro river (Southern and Northern) is

considered a gaining river.

In the northern sector, the Villoresi channel flows (Fig. 1) for about 86 km, from the Ticino (W) to the Adda river (E) feeding a fully gravity-driven irrigation network of secondary and tertiary channels with a total length of about 3 000 km (Fig. 1). The Villoresi channel distributes annually an average volume of about $500 \times 10^6$ m$^3$ of water over an area of about 300 km$^2$. Similarly, the Muzza and the Pavese-Naviglio Grande channels, which flow in the eastern and western sector of the study area,

respectively, feed fully gravity-driven irrigation networks covering about 425 km$^2$ and 463 km$^2$, respectively.

The occurrence of lowland springs (Fig. 1), called *fontanili*, is observed all over the Po Plain in a E-W 20-kilometer-wide belt (about 600-kilometer-long) at the transition between high and low plain. This has been related to the passage from coarse sandy-gravel to medium-fine sand deposits, the consequent decrease in transmissivity and a localized minor change in the topographic gradient. Because of their constant flow rate and temperature (ca. 10 °C to 13 °C) the *fontanili* have been used for

land reclamation, since the XII-XIII century, and for irrigation, since the XVI century, to keep fields unfrozen during winter.

## 2.4 Groundwater withdrawals and levels

Several groundwater monitoring networks collect groundwater data in the study area since the beginning of the XX century. In addition to these, groundwater data were collected on 1994, 2003 and 2014 from over 447 monitoring points in the whole Lombardy region by the regional management and protection water plan (Regione Lombardia, 2006). Pumping rates are

available for 560 wells located in the Milan urban area, whereas for the remaining 1 186 wells they have been estimated by analysing the volume of water distributed for drinking purpose for each municipality (Urban water census: ISTAT, 2013). For the latter, total groundwater abstraction has been assumed to be constant and equal to $246 \times 10^6$ m$^3$/year ($\pm 6\%$) since no significant trend has been found. On the contrary, in the Milan area the abstraction rates changed with time from a maximum





of about $700\times10^6$ m³/year in the middle '70s to about $230\times10^6$ m³/year at present days. In Figure 2, the groundwater level

time-series (1950–2016) are reported for 20 monitoring points in the Milan metropolitan area (Fig. 1) together with the total historical pumping rates. The groundwater levels within the Milan area can be examined with respect to the Italian socio-economic developments. After the post-war reconstruction and during the "economic boom" (1950–1975) the extensive groundwater abstraction caused a decrease of about 10 m to 15 m in the groundwater level. The total abstraction rate in the Milan area reached a maximum of about $700\times10^6$ m³/year. During the oil crisis (1975–1980), the rapid decrease of pumping

rates leaded to a rapid groundwater rise of about 7 m to 8 m. Then, after a stationary period (1980–1985) the decrease of groundwater abstraction restarted following the European Monetary System crisis and the economic decline. Presently, the total abstraction in the Milan area is about $220\times10^6$ m³/year. Consequently, the groundwater levels started to rise since the '90s with a maximum rising rate of about 1 m/year between 2008 and 2014.

The largest oscillations (5 up to 10 m) have been observed in the Milan city area and in the northern sector of the study area,

whereas in the southern sector or in densely cultivated areas (e.g. monitoring points 5, 7 and 11) the water table exhibits only small oscillation (less than 4 m). In this sector, the aquifer system is largely controlled by the hydrographic network that keeps the water table shallow.

## 3 Materials and methods

### 3.1 Multi-dimensional approach for hydrostratigraphic reconstruction

The stratigraphic database (C*ASPITA*, Regione Lombardia, 2016) collects the borehole logs available for the Lombardy-Po Plain area. The database contains information regarding the position, the elevation, the depth and the lithological description of layers crossed by each available borehole or well. 8 628 borehole logs were collected and stored in a georeferenced database (Fig. 1). In addition to these, the 6 high-resolution borehole logs from Regione Lombardia and ENI (2002) were acquired and stored. Then, a multi-dimensional 1D to 3D approach was adopted for the development of the hydrogeological model of the

study area.

### 3.1.1 1D analysis: hierarchical stratigraphy

For purpose of hydrogeologic analysis, it is useful to develop conceptual models to characterize spatial trends in hydraulic conductivity and to predict geometry of hydrogeologic facies from limited field data. After reviewing the literature existing for small subdomains of the study area (Zappa et al., 2006; Comunian et al., 2016, Mele et al., 2010; Cavalli, 2012), a

hierarchical classification of the lithologies was adopted to reclassify them into lithofacies, hydrofacies, and aquifer groups. First, each stratigraphic layer description was codified with an uppercase and a lowercase alphabetical code (Table 1), where the first indicate the prevailing texture (e.g. *G* for gravel, *S* for sand, *M* for silt and *C* for clay), and the latter the dominant grain size dimension (e.g. *c* for coarse, *m* for medium or *f* fine). Compared to previous studies (Zappa et al., 2006; Comunian et al., 2016; Mele et al., 2010; Cavalli, 2012), in this research sedimentary structures such as laminations, cross-beddings and





ripple marks were not taken into account. These local scale sedimentary structures can be neglected during the estimation of the terms of the groundwater budget at the scale of hydrogeological basin (Giudici, 2010). Therefore, the main aquifer and aquitard/aquiclude groups were considered and approximated with equivalent homogenous medium.

Three hierarchical classification levels were adopted:

I. Lithofacies: each class corresponds to defined lithological facies without considering internal texture variations
and textures with limited relevance (less than 5%). The lithofacies were grouped into 13 hydrofacies as function of the predominant texture classes.

II. Hydrofacies: the term is used to indicate interconnected units with relatively homogenous hydraulic properties (Bierkens, 1996). The hydrofacies have a finite horizontal-correlation length, in most cases significantly greater than the vertical-correlation one (Anderson 1989; Anderson et al., 1999; Miall, 1978; Eyles et al., 1983; Keller
200 1996).

III. Aquifer and Aquitard/Aquiclude groups: to obtain a better view of the regional trends for the hydrostratigraphic modelling, the introduction of a further simplification was pivotal. Therefore, the hydrofacies were grouped into 3 aquifer classes (i.e. Unconfined, Semi-Confined and Confined) and 2 Aquitard/Aquiclude.

### 3.1.2 2D analysis: cross-sections and correlation criteria

For the hydrogeological modelling of the aquifer system, a 2-dimensional approach was applied interpreting and correlating one-dimensional data projected on vertical cross-sections. The interpretation of the cross-sections was performed at the hydrofacies level (see Table 1) of the hierarchical classification, and provided the definition of the boundary between hydrofacies, and the basis for the successive interpolation of the surfaces. The interpretation of cross-sections was achieved in three distinct phases to verify the geometric coherence (elevation, intersection and correlation relationship) with geological,
stratigraphic, geochemical information. The three steps are:

I. interpretation of 38 NW-SE (azimuth 152°; see Fig. 1) cross-sections (along flow direction) and 50 E-W (azimuth 61°) cross-sections (perpendicular to flow direction), 2 500m spaced and with a 500m tolerance (i.e. maximum borehole distance from the cross-section trace). Cross-sections containing high-resolution borehole logs (Regione Lombardia and ENI, 2002) were analysed first to highlight the transition between the described sequences in the
adjacent boreholes.

II. interpretation of 26 NW-SE cross-sections and 46 E-W cross-sections. The sections were 1 250m spaced with 300m of tolerance. In addition, geochemical data were added as point data.

III. interpretation of seven variable orientation cross-sections to increase the number of points for 3D interpolation in slightly populated sectors.

At the end of each stage, a manual correction on the control points (i.e. cross-section intersections) was performed.



The cross-sections were interpreted through the analysis of vertical depositional trends (Miall, 1978). The reconstruction of the basal surfaces of the unconfined and semi-confined aquifers was achieved through the individuation of the three main depositional sequences described in par.2.2 (Fig. 3). The lower sequence is strictly related to the meandering river plain depositional processes, hence resulting in a fining upward sequence of medium-fine sand layers interbedded with clay and silt

layers (Fig. 3a). The two-overlaying fining upward sequences correspond to the assemblage of distal and proximal fluvial-glacial outwash facies. The first is composed of sand, gravelly sand, and massive fine clay and silt layers. The overlying one is composed by coarse and medium gravel and sandy gravel. The basal surfaces of the unconfined and semi-confined aquifers correspond to the top of impermeable layers of fining upward sequences, since the transition from clay-silt layers to gravel or sand layers corresponds to erosional surfaces. In practice, the erosional surfaces were defined when a coarse hydrofacies (e.g.

*G*, *GS* or *SG*) overlays a medium grained (e.g. *S*, *SM*, *SC*) or a fine-grained hydrofacies (e.g. *M* or *C*).

Geochemical data provided by local and regional agencies, including 120 655 chemical analyses from 5 075 sampling wells (De Caro et al., 2017) were considered during the interpretation of cross-sections. In particular, the concentration of indicator ion species ($NO_3$, $SO_4$ and $Cl$) was a supplemental interpretation criterion (Fig. 4) adopted in the analysis. In fact, the deep confined aquifer is mostly characterized by natural conditions preserved by the effective separation from the superficial

aquifers (De Caro et al., 2017). On the other hand, pollution affects the shallowest aquifers. Therefore, the concentrations of main indicator ions associated to anthropogenic pollution delineate the effective separation between semi-confined and confined aquifer (i.e. the 25 mg/l contour line of nitrate of Fig. 4).

### 3.1.3 3D analysis: interpolation

The three-dimensional analysis corresponded to the interpolation of the 2D aquifer limits (i.e. points along the interface) to

result in 3D surfaces. The surfaces were interpolated with ArcGIS Geostatistical Analyst tool (Johnston et al., 2001) by using an Ordinary Kriging with trend removal and a smoothing factor to adjust the weights of the neighbourhood points (Table 2). The interpolation was performed for each surface on a training set (80% of the points) and then validated on a test set (20% of the points).

## 4 Groundwater flow models

### 245   4.1 Geometries and mesh

The hydrostratigraphic model was implemented into a 3D finite element model (FEFLOW; Diersch, 2013). The 3D mesh includes 12 040 320 triangular prismatic elements divided in 12 layers (1 003 360 elements per layer). The distance between nodes ranges from 1 500 m down to 10 m in proximity of pumping wells and rivers. The thickness of the 12 layers varies between 3 m and 20 m depending on the thickness of the hydrogeological units and on the well screens position. The thickness



of these layers was assumed constant during the groundwater flow modelling. The vertical discretization can be summarized as follow (Figs. 5a, b):

- Layers 1 to 4: represent the unconfined aquifer. These layers were subdivided (Fig. 5b) according to the distribution of the fan deposits (Fig. 5a) and their internal transition from proximal to distal fringes (gravelly unconfined to sandy unconfined aquifer).

- Layer 5: represents the discontinuous aquitard (3m mean thickness) between the unconfined and the semi-confined aquifers.

- Layers 6 to 11: include the semi-confined aquifer. The thickness (about 10 m) of the sublayers was fixed according to the position of the well screens of the multilayer wells. These layers have been subdivided into distal and proximal sectors (northern and southern sectors, respectively).

- Layer 12: represents the low permeability confined aquifer.

The spatial discretization of the aquifers (i.e. proximal to distal outwash deposits) was verified by the analysis of the ratio between the cumulative thickness ($h_{ydr}$) of each hydrofacies (Fig. 5c) and the total thickness ($h_{tot}$).

## 4.2 Boundary conditions and water budget

Main water inputs and outputs to the hydrogeological system are the recharge at the ground surface, the abstraction of public
supply wells and, the outflow from the lowland springs and rivers. Average monthly recharge rates for the study area were derived from a simplified Penman-Grindley model (Penman, 1950; Grindley, 1970), and the evapotranspiration was calculated using the Thornthwaite's (1948) equation (Fig. 6) from meteorological data (1950-2016) measured in 23 meteorological stations within the Adda-Ticino area (ARPA, 2016). Then, annual recharge values were applied on the model surface by distinguishing urban impervious (no-infiltration) and vegetated areas (Fig. 1). In the urban areas, a recharge rate corresponding
to 15% of the total water supply. Estimates from the two main water suppliers in the metropolitan area (MM SpA and CAPHolding) amount to 10 to 12% of the distributed water. The adopted value is slightly larger than the reported one to include losses from sewer networks which are not estimated by the regional agencies.

The annual recharge values for the Villoresi, the Muzza and the Pavese irrigation areas were estimated by considering the transpiration (i.e. the basal crop coefficient, $K_{cb}$) of the prevalent crop types (maize, cereals and forages; DUSAF, 2012).
Accordingly, the recharge was obtained by scaling the total distributed water volume by the extent of farming areas and by an average basal crop coefficient equal to 0.3 for the Muzza and the Villoresi areas. For the Pavese-Naviglio Grande area a basal crop coefficient of 0.475 was used, since rice is the prevailing crop type (Allen et al., 1998). Average values of about 464 mm/year, 613 mm/year and 850 mm/year were estimated for the Villoresi, the Muzza and the Pavese-Naviglio Grande irrigation areas, respectively (Fig. 1).

Groundwater abstraction from over 1 721 wells (576 in the Milan urban area) was simulated via the Multi-layer wells boundary condition. Accordingly, the wells were simulated with highly conductive one-dimensional finite elements representing the



well pipe. Flow within the well is simulated with the Hagen-Poiseuille cubic law (Diersch, 2013) and the appropriate parameters of the discrete feature are derived from the specified well geometry (i.e. radius and screen position). Most of the wells in the Milan urban area are screened in the semi-confined aquifer, while in other areas they extract water from both the semi-confined and unconfined aquifers.

The eastern, western, and southern boundaries of the model correspond to the Adda, the Ticino, and the Po river, respectively. These boundaries were simulated as a Dirichlet condition based on hydraulic head surveys in 1994, 2003 and 2014 (Regione Lombardia, 2016) Fig. 9). The northern boundary has an imposed head value derived by the interpolated groundwater level maps for 1994, 2003 and 2014 (Regione Lombardia, 2016).

The lowland springs were simulated by assigning a fixed hydraulic head equal to the nodal elevation. With this setting, the fixed-head boundary conditions act as seepage faces (i.e. flux-constrained Dirichlet boundary condition) and an additional constraint condition, that only allows outflow, is applied (Diersch, 2013).

## 4.3 Hydraulic parametrization

Different methods were used to estimate hydraulic parameters. For the unconfined aquifer, grain size distribution data were analysed with different empirical equations to obtain hydraulic conductivity. For the semi-confined aquifer, several aquifer test data were acquired and analysed with proper solutions to obtain transmissivity and conductivity values.

### 4.3.1 Unconfined aquifer

An extensive research literature exists concerning the estimation of hydraulic conductivity of unlithified sediments from grain-size distribution data (Kasenow, 2002). These methods are separated into those that are based on analogies to pipe or capillary flow (Carman, 1937; Kozeny, 1927), and empirical relationships between grain-size distribution and permeability (Alyamani and Sen, 1993; Hazen, 1892; Slichter, 1899). Most of these empirical methods use characteristic values of the size distribution, such as the $d_{10}$ (effective diameter), the $C_u$ (uniformity coefficient), and the $d_{50}$ (median diameter) value. In this study, 113 grain-size distributions of samples collected from the unconfined aquifer (4–40 meters) during the excavation of subway lines in the Milan area were acquired (Fig. 7a). The grain-size distributions cover all the sediment-type spectra (i.e. hydrofacies) and depositional environment within the study area. Therefore, different empirical equations (Alyamani and Sen,1993; Chapuis et al., 2005; Beyer, 1964; Harleman, 1963; Hazen, 1892; Kozeny, 1953; Carman, 1956; Navfac, 1974, from Chesnaux et al., 2011; Sauerbrei, from Vukovic and Soro, 1992; Slitcher, 1899) were applied to estimate the hydraulic conductivity for the specific sediment types (each lithofacies of table 1). Table 3 summarizes the methods used for the estimation of hydraulic conductivity from grain-size distribution. The lithofacies conductivity values were summarized for each hydrofacies of table 1 (Fig. 7a) according to the hierarchical classification principle and to represent the relatively homogenous hydraulic properties of the unconfined aquifer. Then, a representative permeability value ($K_{eq}$) was assigned to each borehole log within the unconfined aquifer by normalizing the conductivity of each crossed layer with the layer thickness, according to:





$$K_{eq} = \sum_{i=1}^{n} \frac{K_i b_i}{b_{tot}} \quad (1)$$

where $b_{tot}$ is the thickness of the aquifer, $b_i$ is the i-layer thickness and $K_i$ is the conductivity value of the i-layer. Then the

equivalent values were interpolated (Fig. 7b) to obtain a map of hydraulic conductivity values.

**4.3.2 Semi-confined aquifer**

The hydraulic parameters of the semi-confined aquifer were derived by analysing 525 well tests (in the Milan city area) and 68 step-drawdown well test (distributed mainly outside the Milan city area). For the 525 well tests only the specific capacity ($S_c=Q/s$) and the well diameter data was available, therefore the Cassan's method (Cassan, 1980) was used to estimate the

transmissivity and the hydraulic conductivity values. The method consists in the evaluation of the σ and θ parameters according to:

$$\sigma = \frac{s}{i \, r_w} \qquad (2)$$

$$\theta = \frac{2\pi s}{Q} T \qquad (3)$$

where $s$ is the drawdown value, $i$ is the hydraulic gradient, $r_w$ is the well radius, $Q$ is the pumping rate and $T$ is the transmissivity.

For each test, the values of $\theta$ were derived from the theoretical curve proposed by Cassan (1980), then the transmissivity values were calculated (Fig. 8a). The Theis solution (1935) with the superposition principle was used for estimating the transmissivity and the hydraulic conductivity values from the step-drawdown well tests. To evaluate the quality of the results (Fig. 8a), the ratio between the estimated transmissivity and specific capacity ($S_c$) values was compared to previous studies that examined the empirical relationships between them (Logan, 1964; Thomasson et al., 1960; Razack and Huntley, 1991; Bakiewicz et al.,

1985). The empirical relationships are based on the simplification of the Thiem equilibrium equation (Thiem, 1906) for which:

$$T = \frac{0.366 \, Q \log\left(\frac{r_1}{r_2}\right)}{(s_1 - s_2)} \, (4)$$

where $Q$ is the abstraction rate (m³/s), $s_1$ and $s_2$ are drawdowns at two different distances, $r_1$ and $r_2$, from the pumping well. Thomasson et al. (1960) simplified the Thiem (1906) equation using theoretical values for the log-ratio term. Combining a mean value for the log-ratio with the other constants, the relationship between $T$ and $S_c$ reduces to the general form:


$$T = A_1 S_c^D \qquad \text{or} \qquad \log T = \log A_1 + D \log(S_c) \qquad (5)$$

with $A_1$ ranging between 0.9 and 1.52. Although the radii are respectively very large and very small, the logarithm of their ratio varies over a small interval. Assuming the quantity $\log(r_1/r_2)$ equal to 3.32 (Rose and Smith, 1957), and a 100% well efficiency, Logan (1964) proposed the following approximation:

$$T = \frac{1.22 \, Q}{s} = 1.22 \, S_c \qquad (6)$$

where, $s$ is the drawdown in the pumping well.





Similarly, Thomasson et al. (1960) and Bakiewicz (1985) proposed values of 1.18 and 1.32 for $A_1$, respectively. Razack and Huntley (1991) proposed an $A_1$ value of 15.3 and a $D$ equal to 0.67. To estimate the parameters of eq. (5) and their confidence intervals, the bootstrap method (Efron, 1982) was used. The distributions of the computed values for the two parameters of eq. (5) are presented in Fig. 8b and c, together with the confidence intervals between the $10^{th}$ and $90^{th}$ percentile.

**4.4 Model calibration**

Steady-state model calibration was performed by inverse procedure (PEST; Doherty et al., 1995). The 2014 mean groundwater levels at 124 selected observation points both in the semi-confined (51) and (73) unconfined aquifers, were used for the calibration. Observation points where groundwater measurements were taken in temporarily turned off wells were excluded. Different values of the anisotropy ratio ($K_v/K_h$) in the range between 0.1 and 0.5 (Todd, 1980) were tested. The estimated
hydraulic conductivity values were used as initial (Table 4) values and adjusted during the calibration. Likewise, the transient-state model (1950–2016) was calibrated by adjusting the specific storage (storativity, $S$) values.

**4.5 Future scenarios**

The calibrated transient-state model was used for simulating future scenarios. The *IPCC RCP4.5* and *RCP8.5* climate projections (for the 2021–2050 period), based on the Special Report on Emissions Scenarios (*SRES*), was considered suitable
for the Milan Po Plain area (Vezzoli et al., 2015). The *RCP4.5* scenario projects a decrease of precipitation during spring and summer (-10% and -21%, respectively), and an increase of mean temperature (0.67 °C to 1.4 °C during winter and summer, respectively). The *RCP8.5* scenario projects an increase of precipitations during winter and autumn (11% and 8%, respectively), and a decrease during spring and summer (-2% and -14%, respectively), with an increase of mean temperature (1.2 °C to 1.4 °C for winter and autumn, respectively). A decrease of about 30% of irrigation was estimated (Gandolfi and
Facchi, 2009) as result of climate change because of a decrease of irrigated surface, increment of winter cereal cropping and of evapotranspiration deficit. Changes in the abstraction rate within Milan area were considered under some demographic change scenario. Projections indicate an increase of about 9% for population in the study area, whereas during the last decade the *per capita* water consumption decreased of about 10% (ISTAT, 2014, 2016) suggesting an asymptotically stabilizing value at about $216 \times 10^6$ m$^3$/year. This is shown in Figure 2 where the groundwater withdrawal data (since 1970) has been fitted with
a second order polynomial function ($R^2$=0.96) up to 2030. Referring to the previously describes *IPCC* scenarios, the following medium-long term future scenarios (2017-2030) were simulated:

    I.    the *RCP4.5* climate projection;

    II.    the *RCP8.5* climate projection;

    III.    the *RCP8.5*-I30P climate projection including a 30% decrease of distributed irrigation water.

These scenarios were simulated assuming that climate change gradually occurs during the 2017–2030 period (i.e. the decrease/increase in precipitation/temperature was linearly spread during the simulation period). Yearly recharge values were



estimated and assigned on the model surface as in section 4.2. For each scenario, pumping rate decreases with time following the assumed future trend. Upper and lower confidence limits of projection curve of future abstraction rate (10 and 90% confidence bands of Figs. 2 and 11b) were simulated as well.

## 5 Results

### 5.1 Hydrostratigraphic reconstruction

The 1-dimensional analysis consisted in grouping the geological data (lithologies) according to several hierarchic orders based on viable stratigraphic and hydrogeologic rules. Within each group, the hydrogeological properties were constant and three hierarchical levels were considered (i.e. lithofacies, hydrofacies, aquifer group). The lithofacies level was the lower hierarchic order and it grouped all the lithologies into 36 classes. Primary and secondary textures were considered (e.g. $G$ groups a where gravel prevails, and $GC$ groups where gravel prevails and clay cannot be neglected). The extreme detail associated with lithofacies description required the introduction of higher classification orders. Thus, the lithofacies were grouped into 16 hydrofacies representing groups with a comparable hydrogeologic behaviour. The upper hierarchical groups were the aquifers (e.g. high-conductivity units, 3 in total) and aquitard/aquiclude (e.g. low-conductivity units, 2 in total).

The 2-dimensional analysis allowed to define the aquifer limiting surfaces (i.e. lines in 2D) which corresponded to regional unconformities. Three fining upward sequences, and their basal surfaces, were distinguished during the interpretation of 121 cross-sections (Figs. 1, 3, and 4). The basal surface of the semi-confined aquifer (Fig. 3) was marked by the transition from alternated silty clayey and sandy layers (e.g. $M$, $SC$, $SM$, $C$) to conglomeratic-sandstone ($R$, in the northern sector) or sandy levels ($SG$, $S$). The basal surface of the overlaying unconfined aquifer was marked by the transition from sandy to gravelly layers ($G$, $GS$) and by a clayey silty aquitard ($C$, $M$).

The 3-dimensional analysis allowed to generate the aquifer limiting surfaces to be used in the groundwater flow model. The Ordinary Kriging with polynomial trend removal and a smoothing factor was used to interpolate the surfaces.

The basal surface elevation of the semi-confined aquifer ranged between 195.1 m a.s.l. and -42.1 m a.s.l. with an average slope of about 0.54 towards SE. The semi-confined aquifer was limited to the top by a discontinuous aquitard with a basal elevation ranging between 162.6 m a.s.l and -1.6 m a.s.l. with an average slope of 0.27 toward SE. The unconfined aquifer bottom elevation ranged between 165.2 m a.s.l and 3.5 m a.s.l. with an average slope of 0.28 toward SE. The model validation accomplished on the test set results in root mean square errors, $E_{RMS}$, between interpolated and measured elevation of 2.75 m, 3.4 m, and 3.23 m, for bottom of semi-confined aquifer, discontinuous aquitard, and unconfined aquifer, respectively.

### 5.2 Hydraulic parametrization

Hydraulic conductivity values for the analysed aquifers were obtained with two different methods. For the unconfined aquifer, equivalent conductivity values resulting from adoption of different empirical equations (Table 3) on 113 grain size distributions





ranged between $5 \times 10^{-2}$ m/s and $1 \times 10^{-4}$ m/s (Fig. 8d). This transition occurs between the high and the low plain, close to the position of the 20km wide lowland spring belt.

Concerning the semi-confined aquifer, analyses of well and step-drawdown (Theis 1935; Cassan, 1960) tests produced comparable results (Fig. 8). The hydrodinamic aquifer parameters vary from high (K=$1 \times 10^{-4}$ m/s; T=$7 \times 10^{-3}$ m²/s) to low values (K=$5 \times 10^{-5}$ m/s; T=$4.5 \times 10^{-3}$ m²/s) moving southward.

The transmissivity–specific capacity, T–$S_c$, relationship (fig. 8a) was studied and compared with those for unconsolidated sediments and heterogeneous alluvial aquifers proposed by other studies (Thomasson et. al., 1960; Logan, 1964; Bakiewicz, 1985; Razack and Huntley, 1991).  Hence, two empirical relationships between the logarithms of $T$ and $S_c$ were found by bootstrapping the eq. (5) parameters (Figs. 8b, c). The following empirical equations were found for well tests eq. (7) and step-drawdown tests eq. (8) datasets, respectively:

$$T = 1.36 S_c^{0.82} \quad \text{or} \quad \log T = 0.31 + 0.82 \log(S_c) \qquad (7)$$

$$T = 1.07 S_c^{0.825} \quad \text{or} \quad \log T = 0.075 + 0.825 \log(S_c) \qquad (8)$$

where both $T$ and $S_c$ are in m²/day.

## 5.3 Groundwater models

The steady-state model was calibrated on the average groundwater head of 2014 computed as the mean of two measurements taken on May and September (mean difference between measured groundwater levels is about 0.25 m), then it was validated on the piezometric levels of 1994 and 2003 (for which only one measurement was available). Scatter plots of differences between observed and computed groundwater levels of steady state models are shown in Fig. 9. Mean residuals are 2.87 m, 3.24 m and 3.47 m, for 2014, 2003 and 1994, respectively. Considering the extent of the study area, calibration results indicated a reasonable agreement between simulated and observed hydraulic heads, and calibrated hydraulic conductivity values lay within the range of the estimated ones (Table 4).

Hydraulic head distributions and flow patterns for the unconfined aquifer in 2014, 2003 and 1994 are shown in Figure 9. Steady-state model results (Fig. 10a, b) indicated relevant groundwater level differences in the Milan metropolitan area, where an increase of about 9 m was observed from 1994 to 2014. Differences in the equipotential field was particularly evident between 2003 and 2014 because of the increase in curvature of the equipotential lines related to the strong increase in water level and the consequent increase in gaining behaviour of the rivers.

The hydrogeological budget as resulting from the steady-state models is shown in figure 10c. Recharge (i.e. precipitation and irrigation) and upstream inflow from mountain basins dominated the inputs to the aquifers with values of about ($0.8 \times 10^9$ to $2.9 \times 10^9$) m³/year and ($3.6 \times 10^9$ up to $6.7 \times 10^9$) m³/year, respectively. Groundwater outflows were represented by gaining rivers, lowland springs, lateral outflow (excluding rivers), and wells abstraction. Estimated lowland springs outflows were about 47 m³/s, 22.4 m³/s and 13.5 m³/s for 2014, the 2003 and the 1994, respectively. Unfortunately, outflow monitoring data for the 236 simulated springs were not available. However, the obtained values agreed with the monitored spring outflow rates of adjacent areas with about the same extent (Bischetti et al., 2012). Estimated outflow rate across the gaining rivers (Lambro,





Adda, Ticino and Po river) ranged between $1.13 \times 10^9$ m$^3$/year and $2.8 \times 10^9$ m$^3$/year (Fig. 10c). The groundwater contributions to stream flows indicatively ranged between 1.5% and 3%, 2.7% and 11% and, 1.3% and 5% of the total peak discharge flow of Ticino, Adda and Po river, respectively. The mean outflow rates across the nodes representing the rivers (Fig. 10d) indicate a mean outflow of about 0.2 m$^3$/s, 0.05 m$^3$/s and 0.04 m$^3$/s for the Adda, the Ticino, and the Po river, respectively.

Groundwater horizontal outflow rate ranged between $2.5 \times 10^9$ m$^3$/year and $5.9 \times 10^9$ m$^3$/year. Vertical flow rates from unconfined to semi-confined aquifer, and from semi-confined to confined aquifer ranged between ($389 \times 10^6$ to $413 \times 10^6$) m$^3$/year, and between ($66 \times 10^6$ to $107 \times 10^6$) m$^3$/year, respectively.

To fit the calculated levels in the unconfined aquifer to the observed values between 1950 and 2016, a transient groundwater flow model was developed. Historical groundwater level time series were available for few monitoring points in the unconfined aquifer (Fig. 1) mainly in the Milano city area. Therefore, this model mainly focused on the impact of changes in pumping rate within the Milan metropolitan area. The transient simulation considered changes in the annual pumping rate, and used the mean annual observed groundwater levels for comparison with the simulated levels. Calibrated specific storage values (using PEST, Doherty et al., 1995) for each subdomain of the model are reported in Table 4, whereas hydraulic conductivity values are maintained the same as from calibrated steady-state model (Table 4).

Initial groundwater levels (on 1950) were taken from the results of the steady-state model excluding 514 abstraction wells designed after 1950. Transient simulation can be roughly distinguished in two periods:

I.   Period of increasing pumping rates (1950–1971): from $190 \times 10^6$ m$^3$/year to $340 \times 10^6$ m$^3$/year. This part of the simulation is affected by major uncertainties (e.g. well completion, starting time and pumping rate, number of available groundwater level measurements per year) and $E_{RMS}$ ranges between 2.5 and 4.2 m (fig. 11a).

II.  Period of decreasing pumping rates (1971–2016): for this period more pumping and observation data are available to increase the confidence in the estimated parameters. The $E_{RMS}$ ranges between 2.5 and 0.9 m (fig. 11a).

Figure 12 summarizes the results obtained for observation points within the Milan metropolitan area for the simulated future scenarios. Groundwater levels of the SC4.5 and SC8.5 scenarios indicated similar trends (Fig. 12b). In the northern sector of the area, a possible increase of about 2.5 m (points 1 and 2) was observed. This increase progressively vanished moving southward where it ranged between 0.4 m and 1 m (points 3, 20 and 21). In proximity of gaining rivers, a slight decrease of about 0.3 m was observed (points 2 and 11). The SC8.5_I30P scenario showed a similar trend, but the maximum groundwater level increase did not exceed 1 m in the unconfined aquifer.

## 6 Discussions

Many cities in their post-industrial stage development are facing a rise of groundwater levels caused by the decrease of abstraction rate (Lerner and Barrett, 1996). Furthermore, global change is expected to affect multiple interrelated factors which can affect the hydrological cycle altering groundwater levels and recharge. In this framework, groundwater reservoir simulation is becoming an extremely dynamic discipline in which sedimentology, stratigraphy, hydro-geochemistry,



hydrogeology, and computational sciences are combined to develop tools for quantifying aquifer dynamics. Furthermore, long-term groundwater monitoring, analysis of complete time series and modelling are fundamental for the understanding of the effects of global changes on groundwater resources.

## 6.1 Hydro-stratigraphic modelling

The geological reconstruction shows how the climatic, tectonic and isostatic forces controlled the stratigraphic settings of the study area by affecting the sedimentation processes. During the Pleistocene, the glacial pulses defined the sedimentation rates and the deposits progradation in the Po Plain. Therefore, the onset of major Alpine glaciations triggered a radical change in drainage patterns and prominent increase in sediment supply (Garzanti et al., 2011). In this context, the multi-dimensional approach for hydro-stratigraphic modelling allows to:

I. recognise and group minor order entities (i.e. lithologies and lithofacies) into hydrogeological entities of higher orders (i.e. hydrofacies and aquifers and aquitard/aquiclude groups). These one-dimensional entities preserve the maximum detail and minor uncertainties in the model, since all the information are contained in the data itself (Cavalli, 2012).

II. recognise and describe the spatial variation of the depositional sequences in the two-dimensional environment (i.e. cross-section). Accordingly, the aquifer limiting surfaces are reconstructed. The two-dimensional analysis implies a loss of details. However, other constrains, such as high-resolution borehole logs, hydro-geochemical point-data, natural background levels of major ions, have been considered during the analyses.

III. recognise and describe the aquifer limiting surfaces in three-dimensional environment to provide the geometric input for groundwater flow numerical simulation.

The produced 3D-hydro-stratigraphy agrees with the conceptual stratigraphic model of the study area. From an hydrostratigraphic point of view, the produced 3D model includes: (i) a deep aquifer made up of a complex sequence of aquicludes and aquifer systems, consisting of interlayered sand and silt/clay layers; this embraces the Aquifer Group C as defined by Regione Lombardia and Eni (2002); (ii) an intermediate aquifer, related to the Aquifer Group B by Regione Lombardia and ENI (2002), consisting of an homogenous sandy-gravel aquifer body; (iii) an upper aquifer with a discontinuous impermeable basal aquitard, missing in the northermost portion of the study area, and corresponding to the Aquifer Group A of Regione Lombardia and ENI (2002). This aquifer is subdivided into 11 subunits representing the different clastic facies of proximal and distal fringes of 5 alluvial fans (Fontana et al., 2014). In the northern sector, the elevation of aquifer limiting surfaces agrees with elevation of major sequence boundaries, which marks widespread progradation of PS2 and PS3 sequences (see 2.2), given by Regione Lombardia and ENI (2002). In the southern sector, major differences are found. In proximity of S. Colombano (Fig. 1), the interpreted sequences boundaries (bases of unconfined and of semi-confined aquifers) are about 20 m to 50 m shallower with respect to the boundary surfaces (base of aquifer group A and B, respectively) proposed by Regione Lombardia and ENI (2002). Most of the high-resolution borehole logs are located in the northern sector of the Milan-Po plain area and different interpretation criteria are used in this research. In any case the subdivision in lithofacies units could allow





the simulation at a much higher detail, which could be considered for more local studies, or could support the generation of
stochastic models.

## 6.2 Hydraulic parametrization and model discretization

The distribution of the hydraulic parameters of the Milan-Po plain aquifers was investigated combining well data logs,
literature permeability values, and geostatistical simulations (Zappa et al., 2006; Bonomi, 2009; Mele et al., 2010; Comunian
et al., 2016). In this research two different approaches are used to estimate hydraulic properties of the aquifers according to
data availability. For the unconfined aquifer, specific empirical equations to predict the hydraulic conductivity for the given
sediment types (i.e. lithofacies) and their grain-size distributions are used. In general, hydraulic conductivity not only depend
on grain size, but it depends on matrix properties (e.g. porosity and specific surface, Bear, 1972). Generally, these parameters
are not defined with grain size-analyses (Vienkene and Dietrich, 2011) and the empirical methods can poorly predict the
measured values with errors ranging over 500 % (Rosas et al., 2014). Nevertheless, the grain-size empirical methods allow to
economically determine the hydraulic conductivity (Cheong et al., 2008).

The estimated hydraulic conductivity values by means of selected empirical equation (firstly applied to lithofacies and then
up-scaled to the hydrofacies of Table 1) are analysed jointly with the distribution of lowland springs and fan deposits (Fontana,
2014) to isolate homogenous sediment bodies (Anderson, 1989). The estimated hydraulic conductivity decreases (Fig. 7b)
towards the springs belt (between high and low plain) and marks the transition between proximal and distal outwash fan
deposits. In the alluvial plain, as the distance increases moving away from the mountains, the river transport capacity decreases
leading to a reduction in the grain size of the deposits (Auge, 2016). Therefore, a gentle-slope surface linking the aquitard
(base of unconfined aquifer) to the ground surface (Fig. 5b) was introduced to separate two grain size domains which result in
a transition between hydrogeologic behaviours. This decrease in transmissivity has long been recognised by means of
hydrological (e.g. occurrence of springs) and geochemical observations (Pilla et al., 2006; De Caro et al., 2017). The ratio
between the cumulative thickness ($h_{ydr}$) of each hydrofacies (Fig. 5c) and the total thickness ($h_{tot}$) for each subdomain (Fig 5,
Table 4) allows to validate the adopted spatial discretization. Proximal sectors show higher percentage of coarse grained
hydrofacies (i.e. $G$, $GS$), while distal sectors show a higher percentage of medium-fine grained hydrofacies (i.e. $S$, $GC$, $SM$).
Similarly, the semi-confined aquifer has been divided into proximal and distal portions, but from the hydrogeological point of
view it shows quite homogenous characteristics.

Internal spatial discretization based on consideration about hydrological and hydraulic parameters distributions is needed for
purpose of hydrogeological modelling. Obtained results suggest that the approach is suitable for defining equivalent
homogenous units and providing a hydraulic conductivity framework in glaciofluvial aquifers within floodplain areas.
Furthermore, aquifer parameters values could be assigned to any chosen hierarchical level (i.e. 36 lithofacies; 16 hydrofacies;
5 aquifer/aquitard/aquiclude), generated from the borehole logs database, by geostatistical methods (i.e. Kriging, Co-Kriging)
or by stochastic simulations (i.e. Sequential Gaussian simulation, Sequential Indicator Simulation, Transition Probability on
categorical variables such as the hydrofacies; Guadagnini et al., 2004; Zappa et al., 2006; Comunian et al., 2011; dell'Arciprete





et al., 2012; Serrano et al., 2014). In their turn, these realisations can be used for local transport problems which generally require a better understanding of heterogeneity than flow problems (de Marsily et al., 2005).

### 6.3 Specific Capacity-Transmissivity relationship

Transmissivity measurements (from step-drawdown tests) are sparse in the study area. In contrast, specific capacity data (well tests) are abundantly available for the Milan metropolitan area. Transmissivity of the semi-confined aquifer is estimated by interpreting well and step-drawdown tests with analytical models (Theis, 1935; Cassan, 1980). The hydraulic conductivity shows typical values for gravelly sandy aquifers. Since numerous studies have been conducted to estimate transmissivity by finding empirical relations between $T$ and specific capacity $S_c$, the bootstrap method is applied to find parameters of eq. (5).

Values equal to 0.82 and 0.825 are found for $D$ (slope of the $T$-$S_c$ log-relationship) for the well tests and the step-drawdown tests datasets, respectively. Similarly, values equal to 1.36 and 1.077 are found for the $A_1$ (intercept of the $T$-$S_c$ log-relationship). Results obtained for the step-drawdown tests are affected by major uncertainties and larger confidence intervals (10th-90th percentiles) with respect to the ones obtained for the well tests (Fig. 8b, c). The obtained values for eq. (5) slightly differ from literature ones (Logan, 1964; Thomasson et al., 1960; Bakiewicz , 1985; Razack and Huntley, 1991). Differences in regression

constants are related to geology, well completions, aquifer heterogeneities, partial penetration of wells, pumping duration, and storage coefficient. As a consequence, slopes differing from unit may indicate that $Sc$ and well efficiency are well correlated (Christensen, 1995; Srivastav, 2007). In practice, the factor 1.22 represents ideal flow conditions and usually it is replaced with a higher value (e.g. 2) to allow for additional well drawdown resulting from partial penetration effects and well losses (Christensen, 1995). The produced empirical relations are similar to those available for heterogeneous sandy and gravelly

sediments (Thomasson et al., 1960; Logan, 1964; Razack and Huntely, 1991). The obtained equations can be easily used for rapid estimations of hydraulic parameters, but a comparison with other standard methods is suggested.

### 6.4 Groundwater models

Late stages of urban development and the relocation of industrial plants outside urban areas result in rising groundwater level. In the Milan metropolitan area, most buildings and underground structures (i.e. subway lines, underground parking and

building foundations) were built during the 1960–1990 period, that corresponds with the period of maximum groundwater abstraction (about $340 \times 10^6$ m$^3$/year). At that time, groundwater levels were not expected to rise and no action was adopted to prevent water inflow and most of the contaminations occurred in the expanded vadose zone. In this framework, quantitative groundwater models allow to predict changes in groundwater levels and the resulting interaction with underground structures and release of contaminants with consequences on water quality.

The calibrated hydraulic conductivity and specific storage values are in good agreement with those of the hydro-stratigraphic model and estimated by means of well-tests and grain-size analyses. Observed and computed groundwater levels show slight differences both for steady and transient model.





For the 1970-2016 period enough well construction dates and observation data are available and allow to increase the level of confidence in the estimated parameters. During this period, the results of the transient model (Fig. 11a) are affected by a low
$E_{\mathrm{RMS}}$ with respect to those of steady state models (Fig. 9).

Groundwater model explains well the historic changes in the piezometric pattern over the study area since the 1950s. During this period, the regional groundwater flow pattern, from recharge (irrigation zone) to discharge areas changed conspicuously. Estimated flow rates for aquifers beneath the Milan urban area (Fig. 11c) suggest that the confining layers (i.e. aquitard) overlying the semi-confined aquifer leaks over the full extent and when the semi-confined aquifer is pumped, groundwater is
withdrawn also from the overlying unconfined aquifer. Budget analysis suggests that the aquitard serves as a water transmitting medium (Fig. 10c). As a consequence, the anthropic imprint deeply affects the groundwater quality of superficial aquifers of the Adda-Ticino area, while the deeper confined aquifer guarantees a good water quality (ARPA, 2015; De Caro et al., 2017). Hydraulic head distribution (Fig. 9) for the unconfined aquifer suggests that groundwater levels heavily depend on distance from discrete features such as wells and gaining rivers. Different dynamic behaviours can be observed. In proximity of
pumping wells (i.e. monitoring points 1, 3, 4, 20 and 21 in fig. 11b) large variations in groundwater levels depending on changes in the pumping rates are observed. Close to gaining rivers (i.e. monitoring points 2 and 11 in fig. 11b) only small oscillations are noticed. The hydrogeological budget suggests that the amount of superficial recharge affects the groundwater flow pattern. For example, during droughty years (e.g. 2003) total spring and river outflows are strongly reduced, whereas during rainy years (e.g. 2014) groundwater levels rise feeding more springs and rivers (Figs. 9, 10d). This sensitivity to
recharge regime could result in strong changes in case of modifications of the climatic regime.

Concerning the differences among the presented models and those previously published for the Adda-Ticino area (Canepa, 2011; Alberti et al., 2016) some specific comments can be given. Previous models are based on the hydrogeological settings given by Regione Lombardia and ENI (2002) and, as presented above, major differences exist concerning the aquifer boundary surfaces. Differences in domain discretisation are mostly related to the use of different modelling codes (i.e. finite element vs.
finite difference). Initial hydraulic conductivity values of previous models are mostly based (Canepa, 2011) on literature data (Bonomi et al., 2014) mainly obtained by literature data, and assigned to the model grid before pilot-point calibration. In this research, a comprehensive hydraulic parametrization supports the equivalent zonally-constant initial hydraulic conductivity values. This parametrization is the result of quantitative grain size analyses differently from other authors which on the contrary rely on qualitative grain size descriptions only. Furthermore, a much larger number of well (525) and step drawdown tests (68)
with respect to other studies (Alberti et al., 2016) was collected at a regional scale (Fig. 8) providing a more spatially distributed and robust description of the hydraulic characteristics of the aquifers. The applied boundary conditions (i.e. recharge and well withdrawals as Neumann condition, rivers and piedmont recharge as Dirichlet) are similar to those applied in other studies with the exception of wells withdrawals and springs which were not included by Canepa (2011) and Alberti et al. (2016), respectively. Therefore, some relevant differences in the groundwater budget are found. Total well withdrawal and recharge
rate values agree with estimates given by Alberti et al. (2016), whereas differences are found for rivers outflows. In fact, despite similar outflow values are estimated for the Ticino, the Lambro and the Po rivers, the outflow value toward the Adda





river is 33% higher with respect to the estimate by Alberti et al. (2016). Lowland spring discharge amount to about 20% of the total outflow. This value strongly differs from the 44% value estimated by Canepa (2011) which could be related to the omission of well withdrawals in the simulation. On the contrary, no spring outflow is considered by Alberti et al. (2016).

## 6.5 Future scenarios

Potential medium-long term effects of climate and withdrawal changes are analysed by simulating three different future *IPCC* scenarios. The obtained results are indicative of mediated conditions on multi-annual basis since the simulated scenarios cannot produce possible enhancement at seasonal scale. All scenarios predict an increase in groundwater levels between 1 and 2 m in the Milan metropolitan area (points 1, 3, 4 in Fig. 12b), whereas in proximity of rivers or irrigated area (points 11 and 20) a slight decrease is observed. Groundwater budget for simulated scenarios (Fig. 12c) shows how the decrease of recharge in highly impervious areas (surface sealing increases of 11% during the 1950–2012 period in northern Italy; Munafò et al., 2015), such as in the Milan city, partly compensate the decrease (-3% respect to 2016) of withdrawals in the Milan area. It is realistic to claim that climate change will have a reduced impact on groundwater resources in the study area in the future decades, assuming that only a minor change will occur in the recharge from the upstream piedmont and prealpine areas. Despite all efforts for mitigate the climate change, in the Milan-Po Plain area specific measures should be foreseen to minimize the effects of the decrease in abstraction rate maximizing the groundwater storage. In fact, groundwater resources could become fundamental in the near future because of global changes and this would require a control on their quality and a protection of their quantity to overcome possible shortages from other sources.

## 7 Conclusions

Developing aquifer models requires a good understanding of sedimentary and hydrogeological processes. In this paper it is demonstrated that a multi-dimensional approach for hydrostratigraphic modelling, together with different methods for estimating aquifer parameters, allow a realistic characterization of glaciofluvial aquifers. The unbalance between groundwater recharge and withdrawals in the Milan urban area led to a groundwater level rise and a quality status deterioration in the last decades. A 3D hydrostratigraphic model of the aquifers beneath the Milan-Po Plain area was constructed based on a hierarchical classification of litho-stratigraphic borehole data. Detailed information on the 3-dimensional architecture and physical attribute of the glaciofluvial sediments was provided considering hydrological features, sedimentation processes and, hydraulic parameters distribution. A 3D numerical model of the whole groundwater system was developed and calibrated both in steady and transient state. The largest component of the regional groundwater system are public supply wells withdrawals, discharge to gaining rivers and springs, recharge from irrigation networks and vegetated areas and, flow transfer between aquifers. The general long-term increase in groundwater level suggests a progressive increase in the base-flow towards the gaining rivers, whereas the difference in hydraulic head between the unconfined and confined aquifer supports the possible contamination of the deep semi-confined aquifers exploited for water withdrawal. Future transient scenarios (2017–2030) and





based on the observation of both decreasing withdrawals and recharge indicate a further increase in groundwater level in the future decades at a lower rate (ca. 0.3 m/year) than in 1970–2016 period.

**8 Acknowledgments**

The authors are deeply indebted to Maurizio Gorla (CAP Holding), Matteo Monti and Marta Gangemi (MM S.p.a) for providing the well and step-drawdown tests. The authors are grateful to Merri Andrea, Valeria Marchese, Nicoletta Dotti (ARPA R.L) and Marina Credali for providing the geochemical and groundwater data (PTUA) and the stratigraphic database (CASPITA), respectively. The Consorzio Est Ticino Villoresi is thanked for making available the irrigation discharge data, 635 and Eng. Fabio Tradigo (ARUP Italy) for the grain size distribution data.

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



**Table 1: hierarchical classification of the lithologies obtained by the 1-dimensional analysis if 8,628 borehole logs data and 6 high-resolution stratigraphies (Regione Lombardia and Eni, 2002).**

| Aquifer[1] and Aquitard/Aquiclude[2] | Depositional environment and age | Lithological composition | Hydrofacies | Lithofacies | Lithologies |
|---|---|---|---|---|---|
| Unconfined[1] | proximal braid plain Upper Pliocene | Gravels with sandy matrix, sandy gravels | *G* | *G* | *cG,mG,Fg,G, GCong* |
| | | | *GS* | *GS* | *GS, cGcS, cGmS, cGS, mGS, fGcS,fGmS, fGS, GcS, GmS, GfS* |
| | | | | *GSC* | *GSC* |
| | | | | *GSM* | *cmGSM, GSM* |
| Semi-Confined[1] | distal braid plain Middle-Lower Pliocene | Sands, sandy gravels | *GC* | *GC* | *cGC, GC, GCS* |
| | | | *GL* | *GM* | *GM* |
| | | | *S* | *S* | *S, cS, cmS, mS, mfS, fS* |
| | | | *SG* | *SG* | *SG, cSG, cmSG, mSG, mfSG, fSG, cSfG, fSfG, ScG, SfG, SGC* |
| | | | | *SGC* | |
| | | | | *SGM* | *cSGM, mSGM, SGM* |
| | | | *SM* | *SMG* | *cSMG, cmSMG, mfSMG, fSMG* |
| | | Conglomerates, sandstones | | *Ar* | *Ar, ArS* |
| | | | *R* | *Cong* | *Cong, CongC, CongAr, CongG, CongGC, CongS, CongSC* |
| | | | | *R* | *R* |
| Confined[1] | Continental-marine transition Lower Pliocene - Miocene | Sandy lenses within clay and silt layers | *M* | *M* | *M* |
| | | | | *MC* | *MC, MCS, MCG* |
| | | | | *MS* | *MS, MSC, MSG* |
| | | | *SC* | *SC* | *cSC, mSC, fSC, SC* |
| | | | *SM* | *SM* | *fSM, mSM, SM* |
| | | | | *SMC* | *fSMC, SMC* |
| | | | *C* | *CM* | *CM, CMG, CMS, CMP* |
| | | | | *CS* | *CS, CSG, CSM, CSP* |
| Aquitard[2] | 0.45 Ma | Silty layers, silty sands | *M* | *M* | *M* |
| | | | | *MC* | *MC, MCS, MCG* |
| | | | | *MG* | *MG* |
| | | | | *MS* | *MS, MSC, MSG* |
| | | | *SC* | *SC* | *cSC, mSC, fSC, SC* |
| | | | | *SCG* | *SCG* |
| | | | *SM* | *SM* | *fSM, mSM, SM* |
| | | | | *SMC* | *fSMC, SMC* |
| | | | | *SMG* | *cSMG, cmSMG, mfSMG, fSMG* |
| Aquiclude[2] | 0.87 Ma | Clayey and silty layers | *C* | *C* | *C, CCr, CCong* |
| | | | | *CG* | *CcG, CG, CGS* |
| | | | | *CM* | *CM, CMG, CMS, CMP* |
| | | | | *CS* | *CS, CSG, CSM, CSP* |
| | | | | *CP* | *CP* |

| **Alphabetical codification** | | | | | |
|---|---|---|---|---|---|
| **Prevailing textural codes** | | | | **Grain size** | |
| *G* | Gravel | *Cong* | Conglomerate | *c* | coarse |
| *S* | Sand | *M* | Silt | *cm* | coarse-medium |
| *R* | Rock | *C* | Clay | *m* | medium |
| *Ar* | Sandstone | *P* | Peat | *mf* | medium-fine |
| | | | | *f* | fine |



**Table 2: summary of ordinary kriging parameters used for the 3D interpolation of limiting aquifer surfaces**

| Surface | Unconfined bottom | Semi-confined top | Semi-confined bottom |
|---|---|---|---|
| Trend removal order | 1st order | 1st order | 1st order |
| Searching neighborhood | Smooth | Smooth | Smooth |
| Smoothing factor | 0.2 | 0.4 | 0.2 |
| Semiaxis (m) | 44 188.9 | 44 914.6 | 38 097.1 |
| Variogram | Semi-variogram | Semi-variogram | Semi-variogram |
| Variogram type | Stable | Stable | Stable |
| Nugget (m) | 5.29 | 10.17 | 9.65 |
| Range (m) | 44 188.8 | 41 914.6 | 38 097.1 |
| Sill (m) | 549.5 | 604.34 | 172.73 |










**Table 3: empirical equations used to estimate the hydraulic conductivity from grain size distributions according to the adopted criteria**

| Method | Equation (K m/s) | Criteria | Hydrofacies |
|---|---|---|---|
| Alyamani and Sen (1993) | $K = \beta[I_0 + 0.025(d_{50} - d_{10})]^2$ | Well-distributed sample | G, GS |
| Chapuis et al. (2005) | $K = \beta\left(\dfrac{d_{10}^2 e^3}{1 + e}\right)^{0.7825}$ | $0.03 < d_{10} < 3$ mm | GC, GM, S, SG, SM |
| Beyer (1964) | $K = \beta\dfrac{g}{\nu}\log\dfrac{500}{C_u}d_{10}^2$ | $0.06 < d_{10} < 0.6$ mm | GM, S, GC, SM, SG |
| Harleman (1963) | $K = \beta\dfrac{\rho g}{\mu}d_{10}^2$ | | all |
| Hazen (1892) | $K = \beta\dfrac{g}{\nu}[1 + 10(n - 0.26)]d_{10}^2$ | $0.1 < d_{10} < 3$ mm and $C_u < 5$ | G, GS, SG, SM |
| Kozeny (1953) | $K = \beta\dfrac{g}{\nu}\dfrac{n^3}{(1 - n)^2}d_{10}^2$ | Coarse grain sands | S, SG, SM |
| Carman (1956) | $K = \beta\dfrac{\rho g}{\mu}\dfrac{n^3}{(1 - n)^2}d_{10}^2$ | Silts, sands and gravelly sands | M, SC, SM, C |
| NAVFAC (1974; from Chesnaux et al., 2011) | $K = \beta 10^{1.291e - 0.6435}d_{10}^{10^{0.5504 - 0.2937e}}$ | $0.1 < d_{10} < 2$ mm | SG, SM, S, SC |
| Sauerbrei (from Vukovic and Soro, 1992) | $K = \beta\dfrac{g}{\nu}\dfrac{n^3}{(1 - n)^2}d_{17}^2$ | Sands and sandy clays | S, SC, SM |
| Slichter (1899) | $K = \beta\dfrac{g}{\nu}n^{3.287}d_{10}^2$ | $0.01 < d_{10} < 5$ mm | S, SG, GS |

*Water Temperature = 25°C; $\mu = 8.89e^{-4}$ Pa s; $\rho = 999.075$ kg/m³; $\nu = 0.8902$ mm²/s




**Table 4: estimated hydraulic conductivity of unconfined and semi-confined aquifers used as initial values for the groundwater flow model (for zone location and extent see Fig. 5) and, calibrated values of hydraulic conductivity and specific storage.**

| | Aquifer/Zone id | Estimated K [m/s] Min | Estimated K [m/s] Mean | Estimated K [m/s] Max | *Calibrated $K_h$ [m/s] | Calibrated $S_s$ [$m^{-1}$] |
|---|---|---|---|---|---|---|
| Unconfined aquifer[1] | *Molgora mf - Proximal* | $1.63\times10^{-3}$ | $6.56\times10^{-3}$ | $1.15\times10^{-2}$ | $3.35\times10^{-2}$ | $1.71\times10^{-4}$ |
| | *Molgora mf - Distal* | $1.34\times10^{-3}$ | $5.60\times10^{-3}$ | $9.86\times10^{-3}$ | $1.41\times10^{-3}$ | $5.26\times10^{-5}$ |
| | *Lambro mf - Proximal* | $1.67\times10^{-3}$ | $6.77\times10^{-3}$ | $1.19\times10^{-2}$ | $7.93\times10^{-4}$ | $3.04\times10^{-4}$ |
| | *Lambro mf - Distal* | $2.03\times10^{-3}$ | $8.30\times10^{-3}$ | $1.46\times10^{-2}$ | $1.94\times10^{-3}$ | $1.22\times10^{-4}$ |
| | *Seveso f - Proximal* | $1.33\times10^{-3}$ | $5.54\times10^{-3}$ | $9.75\times10^{-3}$ | $2.25\times10^{-3}$ | $5.17\times10^{-4}$ |
| | *Seveso f - Distal* | $1.57\times10^{-3}$ | $6.45\times10^{-3}$ | $1.13\times10^{-2}$ | $2.48\times10^{-3}$ | $1.25\times10^{-4}$ |
| | *Lura f - Proximal* | $1.54\times10^{-3}$ | $6.22\times10^{-3}$ | $1.09\times10^{-2}$ | $3.85\times10^{-4}$ | $9.54\times10^{-5}$ |
| | *Lura f - Distal* | $2.11\times10^{-3}$ | $8.50\times10^{-3}$ | $1.49\times10^{-2}$ | $3.36\times10^{-3}$ | $1.84\times10^{-5}$ |
| | *Olona mf - Proximal* | $1.80\times10^{-3}$ | $7.25\times10^{-3}$ | $1.27\times10^{-2}$ | $6.70\times10^{-3}$ | $2.82\times10^{-4}$ |
| | *Olona mf - Distal* | $1.52\times10^{-3}$ | $6.26\times10^{-3}$ | $1.10\times10^{-2}$ | $1.96\times10^{-3}$ | $5.70\times10^{-4}$ |
| | *Post-LGM* | $9.99\times10^{-4}$ | $4.27\times10^{-3}$ | $7.54\times10^{-3}$ | $2.35\times10^{-3}$ | $5.91\times10^{-5}$ |
| Low-permeability layers[1] | *Aquitard* | $8.38\times10^{-9}$ | $3.02\times10^{-5}$ | $6.03\times10^{-5}$ | $8.61\times10^{-7}$ | $2.56\times10^{-3}$ |
| | *Aquiclude/Confined aquifer* | $1.00\times10^{-9}$ | $5.50\times10^{-9}$ | $1.00\times10^{-8}$ | $1.00\times10^{-7}$ | $2.03\times10^{-2}$ |
| Semi-Confined aquifer[2] | | **Estimated K [m/s]** | **T [$m^2$/s]** | | | |
| | Semi-Confined proximal | $1.36\times10^{-4}$ | $7.30\times10^{-3}$ | | $5.51\times10^{-5}$ | $9.60\times10^{-5}$ |
| | Semi-Confined distal | $7.87\times10^{-5}$ | $4.73\times10^{-3}$ | | $7.66\times10^{-5}$ | $1.61\times10^{-4}$ |

[1] *K from grain-size distributions*
[2] *K from pumping tests*
* *Kv = 0.5 Kh*





**Figure 1: Map of the study area showing the boreholes/wells with available stratigraphic/lithologic logs, the hydrological network (i.e. rivers, springs and irrigation channels), the grid of 2D cross-sections (121), the distribution of vegetated and impervious areas, and the distribution of groundwater levels monitoring points.**



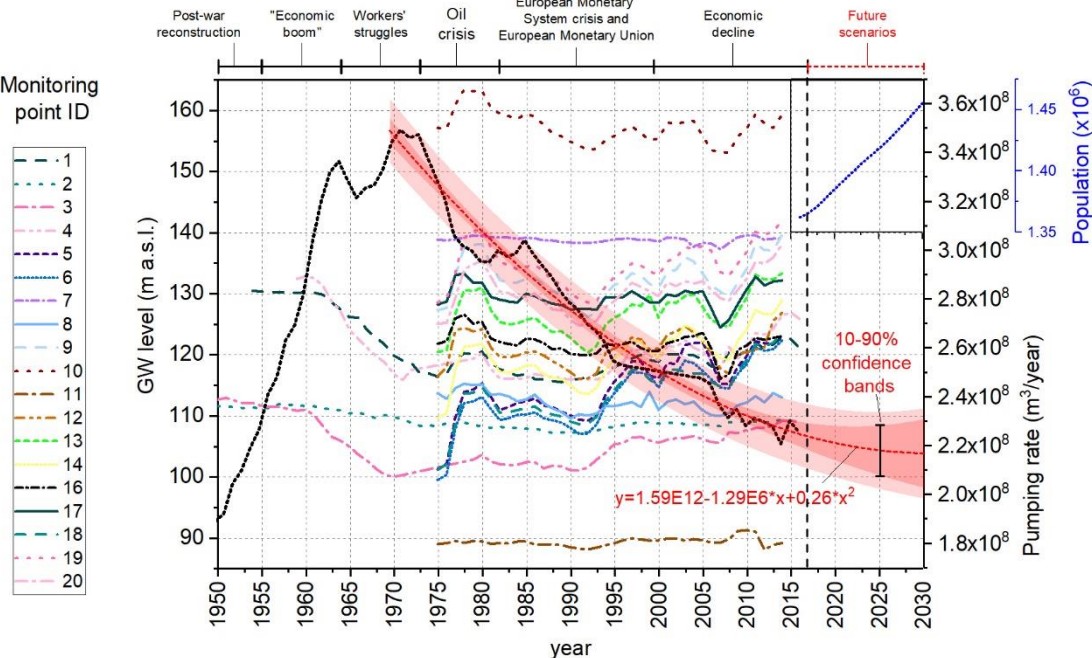


**Figure 2: Monitoring data of groundwater levels for the unconfined aquifer (see Fig. 1 for the points position), total groundwater abstraction rate for the Milan area (dashed line) and polynomial fit (red dashed line) used for future scenarios simulation. Projection of population growth between 2017-2030 is reported in the upper-right box.**



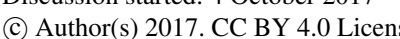

**Figure 3: Stratigraphic reconstruction (a) strip-logs of the high resolution-borehole logs (Regione Lombardia and ENI, 2002) showing the hydrofacies vertical distribution and the fining upward sequences (see lower map for borehole position) and, (b) example of a N-S interpreted cross-section.**



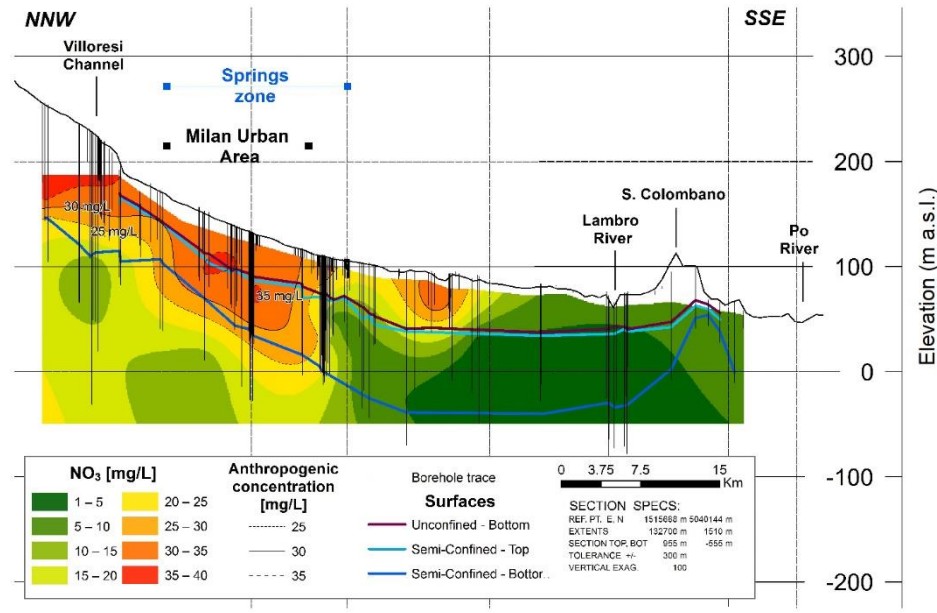

**Figure 4: Example of a N-S interpreted cross-section including distribution of nitrate concentration. Concentration contours (mg/L)**
**refer to values beyond the natural background level (NBL) values as determined via component separation analysis (De Caro et al., 2017) and can be attributed to anthropogenic contamination.**





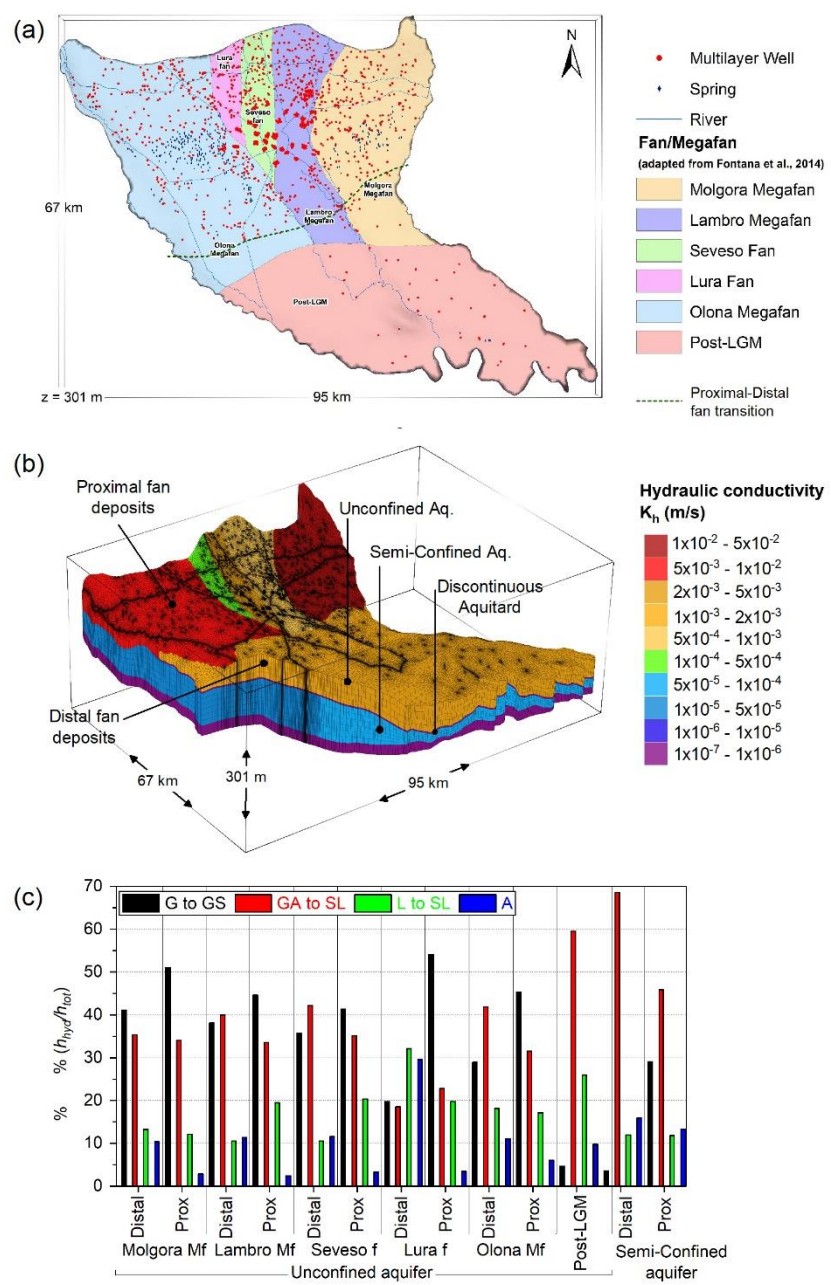

**Figure 5: Spatial discretization of the 3D numerical model (a)** Plan view of the 3D groundwater model showing the horizontal discretization according to the fan/megafan distribution (adapted from Fontana et al., 2014); **(b)** 3D view of the groundwater flow model showing the vertical discretization and, **(c)** percentage distribution of each hydrofacies (see Table 1) within each zone of the 3D hydrogeological model.





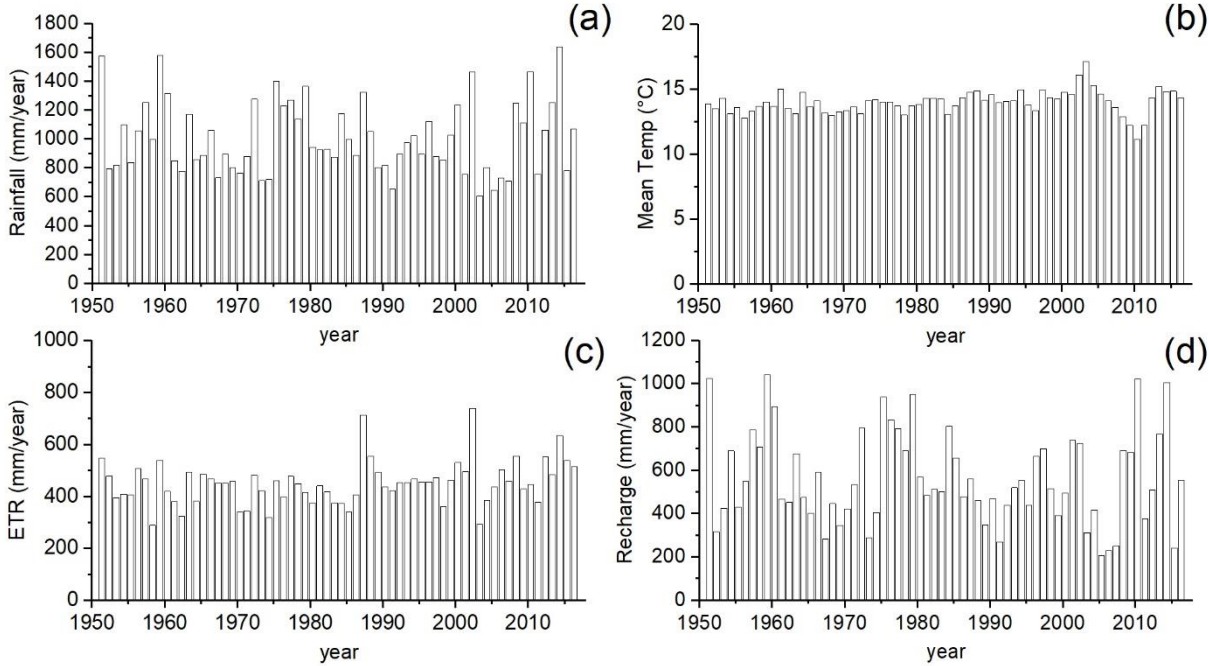

**Figure 6: Bar plots of meteorological data (1950 to 2016). (a) total annual rainfall, (b) mean annual temperature, (c) total annual evapotranspiration (by Thornthwaite's eq.), and (d) annual infiltration values resulting from the hydrological budget.**





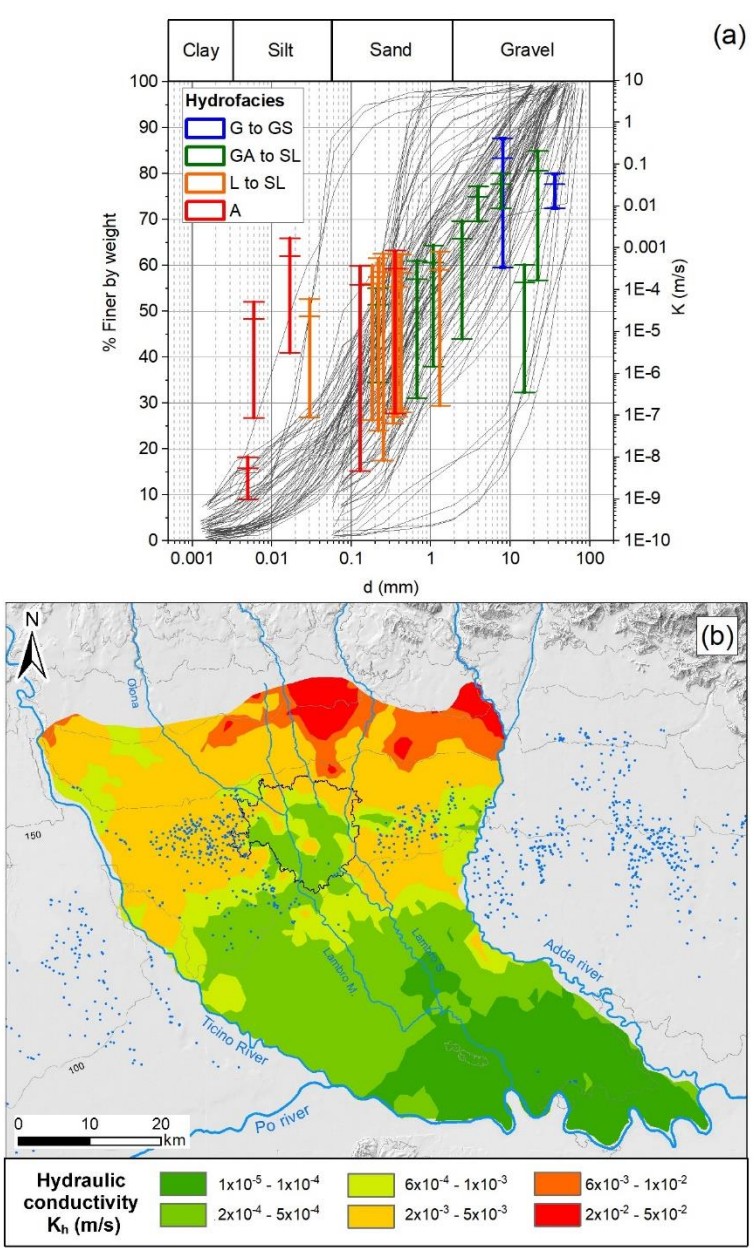


**Figure 7: Hydraulic characterization of the unconfined aquifer: (a) Grain size distributions (113) used for estimating the hydraulic conductivity; vertical bars show the minimum, the median and the maximum hydraulic conductivity values for each hydrofacies of Table 1, and (b) map of equivalent hydraulic conductivity for the unconfined aquifer.**





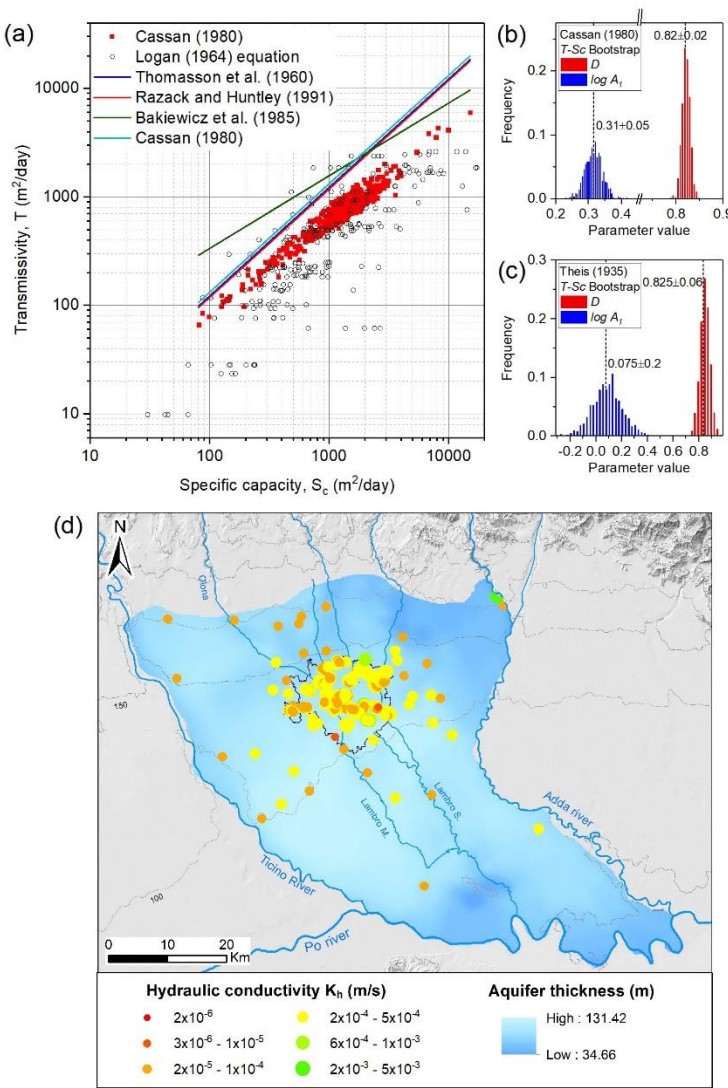


**Figure 8: Hydraulic characterization of the semi-confined/confined aquifer: (a) Scatter plot of transmissivity vs specific capacity values estimated by well and step-drawdown tests and compared to empirical relationship between T and Sc; (b) and (c) bar plots of bootstrap realisations for deriving equation (2) parameters (see text) for the well test and the step-drawdown tests dataset, respectively; (d) map of estimated hydraulic conductivity values; the thickness of the semi-confined aquifer is reported.**






**Figure 9: hydraulic heads flow patterns and scatter plots (observed vs. simulated) resulting from steady-state groundwater models for (a) 2014, (b) 2003, and (c) 1994.**



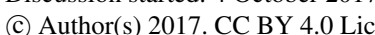


Figure 10: (a) NS and cross-sections of simulated steady-state groundwater levels (for sections location see Fig.1), (b) NS cross section of simulated steady-state groundwater levels within Milan area; (c) scheme of rate-budget for the simulated aquifers showing minimum and maximum inflow, outflow, and aquifer transfer rates. Minimum and maximum values refer to 2003 and 2014, respectively (droughty and rainy year) and, (d) frequency distributions of outflow rate (m³/s) across river (Adda, Ticino and Po) boundaries.



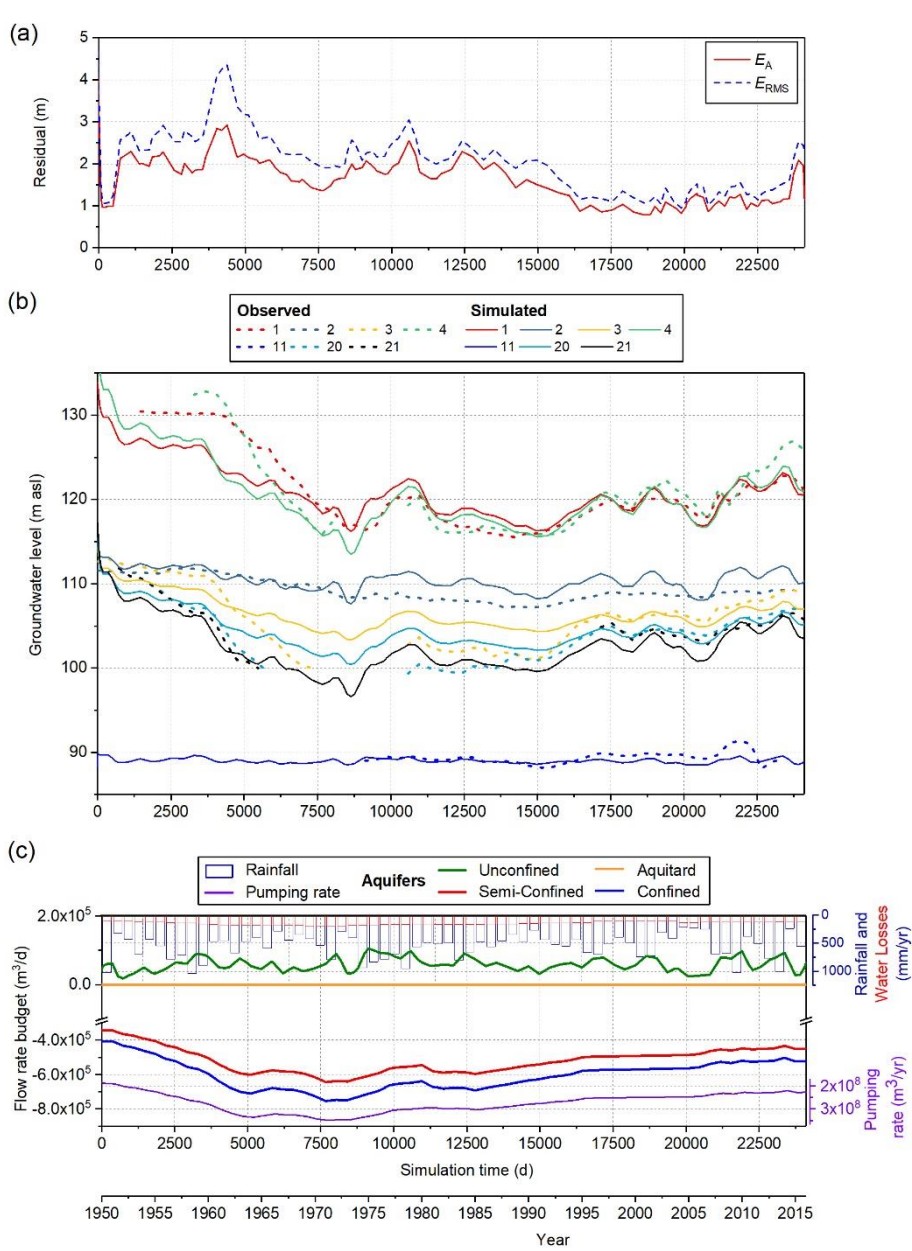


**Figure 11: (a) residuals of transient simulation, solid and dashed line are the absolute and the root mean square errors; (b) results of transient simulation for 1950-2016 period; solid and dashed lines are the simulated and the observed groundwater levels at selected points within Milan area, respectively and, (c) flow rate budget (m³/d) for each simulated aquifer within the Milan area. Pumping and recharge rates are reported as well.**



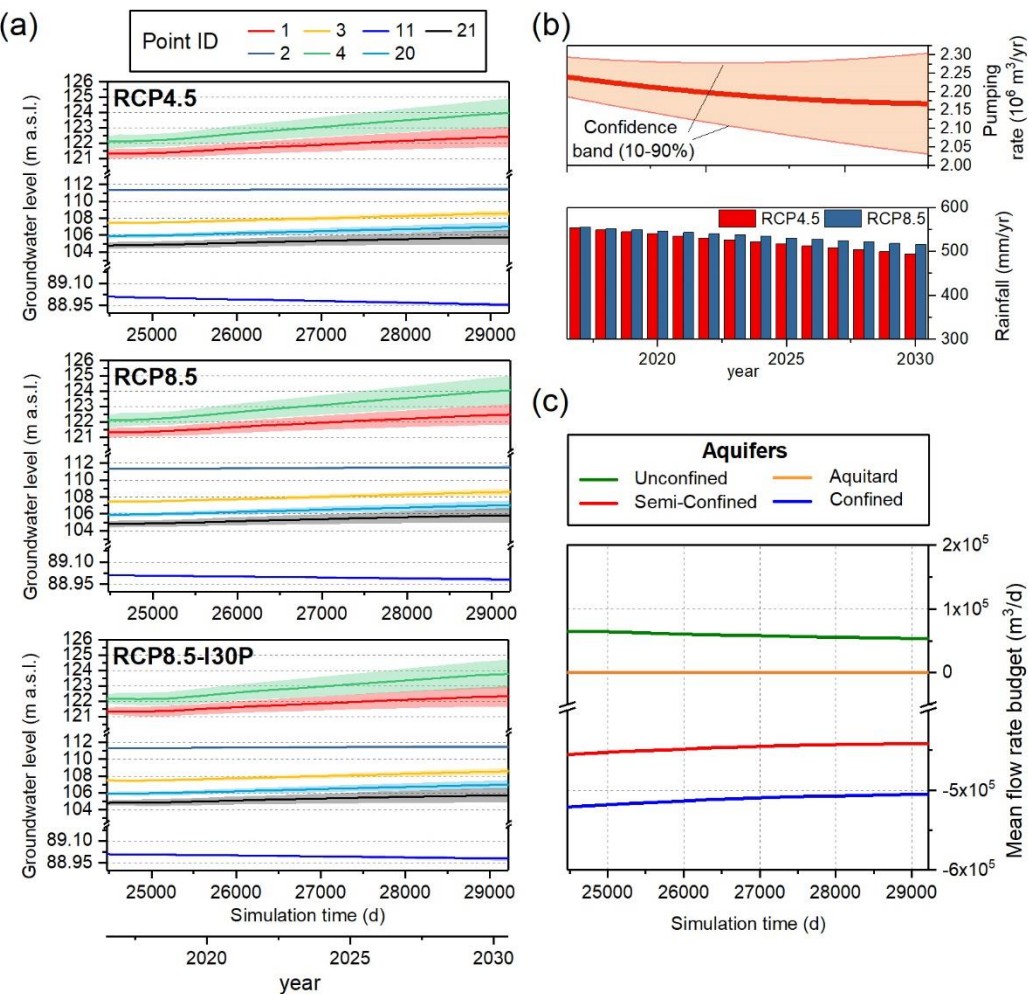


**Figure 12: Results of transient simulations of future IPCC climatic scenarios (2017-2030) and changes in groundwater abstraction, for selected points in the Milan metropolitan area (see Fig.1 for location): (a) projections of future groundwater abstraction scenario and rainfall recharge according to the IPCC (2008) climate scenarios; (b) results for the unconfined aquifer groundwater level, with transparent colour bands corresponding to upper and lower confidence bands for expected future abstraction rates; (c) mean flow**
**rate budget (m³/d) computed within the metropolitan Milan area for each aquifer type. Mean values are computed by averaging the results of the different simulated scenarios.**