# Peer review of "Hydrofacies reconstruction of glaciofluvial aquifers and groundwater flow modelling in a densely urbanized area under changing climatic conditions"

_Hydrology and Earth System Sciences, 2017_

## Referee Comment (RC1) · Anonymous Referee #1 · 4 Dec 2017

This paper shows a comprehensive investigation of Milan – Po Plain aquifers in Italy, including detailed geology and stratigraphy, hydrochemistry, human factors, etc. The authors did a lot of field works, collected many data and conducted a huge hydrogeological model. However, in my opinion, the manuscript needs to be improved to be published on HESS. The paper structure is well-organized, but some sections remain unclear. The introduction is vague and too focused on general aspects. The structure of this manuscript is more like a hydrogeological survey report or groundwater modeling summary, not a research article. Motivation is not clearly stated and the objectives

are vague and of weak scientific interest. Why did you do this study and what scientific questions are answered? Note that the objectives of this paper are differently stated in the abstract and in the Introduction. I recommend fixing the main significant scientific objective, as in my opinion, the multi-dimensional approach used for the reconstruction of aquifer geometry. The objectives and conclusions do not represent a relevant advance to scientific knowledge related hydrogeological characterization. I encourage the authors to address the scientific issues, restructure the manuscript and resubmit the paper.

Because it is a research article instead of a report, the authors are expected to explain why Milan – Po Plain aquifers are important and interesting to study in a scientific point of view, not only for the need of groundwater managing purposes. These aspects have to be clarified and illustrated in the manuscript. The authors used a great proportion of the words in this manuscript to introduce and describe the field works and data collections, data treatment, conceptual model description and model results. Again, I would recommend the authors to focus on the discussion of main significant scientific objective, i.e. the reconstruction of aquifer geometry and how this can be assessed by means of groundwater modeling. The Results and Discussion sections are too long and show repeated information and the resulting redaction is unclear. These sections have to be rewritten and shortened. The Discussion Section is a description of how the numerical model was constructed and performed, and how this is supported by the literature. The rest is a general overview of the results of the model versus measurements (heads, river outflows, recharge . . .) and also the result of the future scenarios simulations. Section 4.4 (Model Calibration) is weak. No relevant comments about the methodology used for calibration. No relevant comments are focused to justify the obtained model calibration statistics. No comparison between the initial (conceptual model) and calibrated parameters (K, Ss, Recharge, etc.). No comparison between initial groundwater budget derived from conceptual model statement (in the whole or in a certain period) respect the obtained by modeling. Some of these issues are weakly illustrated in some parts of the text and in some figures, but I consider that an integrative and detailed explanation is needed. No sensitivity analysis of the parameters used in the model is commented. It is interesting to state any trends of hydrochemistry data variation along time, related to changes in land uses and anthropogenic factors. And a specific discussion of the effect of contamination/pollution/human factors to data is desired. Authors have to explain how the reconstruction of aquifer geometry can be assessed by means of groundwater modeling and justified by means of a sensitivity analysis. Conclusions are too general and obvious. The authors do not need to mention the results of the numerical model in the conclusion part. Authors could use the last section of the paper to extend their discussion to make clear the new contributions that the manuscript supports. Include important implications of your work on the scientific community and also for local groundwater management.
* * *

---

## Author Comment (AC1) · 15 Dec 2017

RC: This paper shows a comprehensive investigation of Milan – Po Plain aquifers in Italy, including detailed geology and stratigraphy, hydrochemistry, human factors, etc. The authors did a lot of field works, collected many data and conducted a huge hydrogeological model. However, in my opinion, the manuscript needs to be improved to be published on HESS. The paper structure is well-organized, but some sections remain unclear.

**AC: We thank the reviewer for this general statement and comments.**

RC: The introduction is vague and too focused on general aspects.

**AC: We believe that the introduction should provide a background and an analysis of the state of the art to identify existing shortcomings of the subject matter. In particular, the paper discusses the relevance of a robust investigation and modelling of aquifers in urban areas. Because of the specificities of the setting, we believe it is appropriate to introduce general problems and the different approaches adopted in the literature. This is addressed as follows:**

**Lines 27-34 provide background information about groundwater rebound resulting from decreasing industrial withdrawals in urban areas, summarizing the major consequences and providing examples.**

**Lines 43-50 describe how climate changes may alter the hydrological cycle. We believe that this is particularly relevant because potential impacts of global climate changes on the groundwater are largely unknown, especially for densely populated areas where groundwater is heavily exploited for public and industrial supply. Available studies about the impacts of climate changes on groundwater resources show site-specific results and are cited.**

**Lines 51-59 explain why an approach employing all the available data, in order to develop a comprehensive hydrostratigraphic model based on an exhaustive parametrization, and to build a robust hydrogeological numerical model, is needed.**

**Lines 60 - 73 introduce the reader to the study area (i.e. the Milan Metropolitan Area) and explain why studying it is within the above mentioned scopes. The state of the art about the study area and what is the original contributions of the research with respect to the existing literature are clearly stated.**

RC: The structure of this manuscript is more like a hydrogeological survey report or groundwater modeling summary, not a research article.

**AC: We believe that the presented manuscript follows the typical structure of a scientific paper, including: i) an introduction to the problem, ii) the presentation of the settings, the data and the methods, iii) a summary of the results, iv) a thorough discussion showing advancements and improvements with respect to the existing literature, and v) the conclusions.**

RC: Motivation is not clearly stated and the objectives are vague and of weak scientific interest. Why did you do this study and what scientific questions are answered?

I recommend fixing the main significant scientific objective, as in my opinion, the multi-dimensional approach used for the reconstruction of aquifer geometry.

**AC: In our opinion, the aims of the research are clearly stated in the abstract and in the introduction, as well as the motivation for this research**

**The groundwater rebound in urban/agricultural/industrial areas is a serious problem in many cases, and may be boosted/buffered by climate change and consequent changes in urban and societal development.**

**To complete a robust analysis: (1) the aquifer geometry needs to be reconstructed, then (2) a reliable groundwater flow model based on comprehensive aquifer parametrization must be developed. The latter will allow to understand the present day conditions and to simulate future scenarios, taking into account climate change, population growth, and industrial and agricultural activities. The importance of such future-scenario simulations is self-explaining, in our opinion.**

**Building such models (i.e. hydrostratigraphic and flow model) for any alluvial aquifer requires the description of the aquifer heterogeneities and the spatial distribution of hydraulic parameters (lines 54-57). These analyses generally involve detailed data and significant computational efforts, as required, for example, by stochastic simulations. Thus, the proposed approach which employs available data to estimate and assign hydraulic parameters over different "homogenous" units of a 3D numerical model can be useful for these kind of analyses. A sedimentological analysis of the stratigraphic data is required to avoid simplified and unreliable approaches and to help grouping lithological layers in significant hydrostratigraphic units. The attribution of hydraulic parameters following the qualitative description of single layers as from the drilling phases is often affected by the subjectivity in the description. Recognition of specific sedimentological and stratigraphic sequences adds meaning and robustness to this work. We present a complete approach that can be followed and replicated by other researchers in this and in similar areas. Therefore, we consider this an important scientific point addressed in this manuscript.**

RC: Note that the objectives of this paper are differently stated in the abstract and in the Introduction.

**AC: We agree that the objectives of the paper are structured in different way in the abstract and in the introduction. For sake of simplicity, we decided to modify the objectives in the introduction to fit the corresponding aims in the abstract.**

RC: The objectives and conclusions do not represent a relevant advance to scientific knowledge related hydrogeological characterization. I encourage the authors to address the scientific issues, restructure the manuscript and resubmit the paper. Because it is a research article instead of a report, the authors are expected to explain why Milan - Po Plain aquifers are important and interesting to study in a scientific point of view, not only for the need of groundwater managing purposes. These aspects have to be clarified and illustrated in the manuscript.

**AC: In the following we try to answer to the reviewer questions but it is possible that, because of language problems, we do not fully understand his/her observations and comments.**

**Since the importance of studying the Milan – Po plain aquifer (and in general alluvial aquifers beneath large cities affected by groundwater rebound) seems to be somehow hidden in the paper, some aspects are not better explained in the introduction.**

**Rising groundwater levels in large cities due to a decline in groundwater exploitation is a serious problem with significant social, environmental and economic implications, and glacial outwash deposits are important sources of groundwater in various areas surrounding alpine chains. This is quite relevant for the areas around the European Alps where very large cities developed with an enormous social and economic value. Accordingly, we believe that the proposed approach could be adopted for designing models aimed at regional groundwater management purposes.**

**The methods and general approach are illustrated for the aquifers of the Milan Metropolitan area. These represent an important groundwater resource for a very large population (ca. 5 million people, half the population of Lombardy region) and a highly industrialized area (listed as the fourth European city for GDP). Nevertheless, a groundwater flow model capable to capture the overall groundwater dynamics (both in steady and transient state) based on a comprehensive aquifer reconstruction and parametrization is still lacking (lines 66-74).**

RC: The authors used a great proportion of the words in this manuscript to introduce and describe the field works and data collections, data treatment, conceptual model description and model results.

**AC: Sorry we do not understand this statement. Field work is not described in the manuscript. On the contrary, we clearly present the available and used datasets and explain all the methods to make the approach understandable as well replicable, as requested for a scientific study.**

RC: Again, I would recommend the authors to focus on the discussion of main significant scientific objective, i.e. the reconstruction of aquifer geometry and how this can be assessed by means of groundwater modeling.

**AC: The discussions, which is quite a relevant part of the paper, is focused on the different objectives of the research, as listed in the introduction: hydrostratigraphic modeling (lines 470-500), hydraulic**

**parametrization and groundwater model design (lines 501-551), groundwater flow modeling and processes identification (lines 552-599), and finally future scenarios (lines 600-613). For each of these topics, we clearly state the problems, advantages and disadvantages of the methods and discuss the results.**

**In our opinion, the exhaustive hydraulic parametrization and its implementation into the 3D finite element model, the comprehensive budget analysis, and the simulations of future scenarios are significant to accomplish the established scientific objectives including the validation of a method for the hydrostratigraphic reconstruction and the groundwater model setting.**

RC: The Results and Discussion sections are too long and show repeated information and the resulting redaction is unclear. These sections have to be rewritten and shortened.

**AC: We will try to clean the discussion by deleting possible duplications. We believe that the results section is clear and concise, whereas the discussions section explores the significance of the results of the work and analyses the significance of the findings in terms of paper topics, approaches and methods.**

**For example, section 5.3 (results) describes the numerical results of the groundwater flow model. First describes the quality of the steady-state numerical model in terms of residual errors (lines 416-423), the validation results (lines 423-426), and the resulting budget (lines 428-441). Then, in similar way the results of transient numerical model and future scenarios are described (lines 442-461).**

**In section 6.4 (discussions), the significance of the modelling results is stressed in relation to paper topics (i.e. groundwater rebound, effect of withdrawals and recharge changes).**

RC: The Discussion Section is a description of how the numerical model was constructed and performed, and how this is supported by the literature. The rest is a general overview of the results of the model versus measurements (heads, river outflows, recharge . . .) and also the result of the future scenarios simulations.

**AC: The model construction is reported in the methods not in the discussion. In our opinion, the discussion section explores all the produced outcomes from the hydrostratigraphic reconstruction to the future scenario simulations results, not only the model construction. Each discussions subsection is more speculative with respect to the results subsections. Anyway, as said above, some parts will be shortened to make the text crispy and keep the focus on discussing the results and methods, as well as the differences with respect to previous studies.**

**Section 6.1 describes how the multi-dimensional approach allows the aquifer geometry reconstruction. The differences with previous hydrogeological models are stressed.**

**Section 6.2 describes both advantages and limitation of methodologies used for the aquifer parametrization. Then the section explains how the results can be implanted into the 3D groundwater model to define nearly-homogenous subunits.**

**Finally, suggestions for the adoption of the hierarchical classification jointly with the hydraulic parametrization (i.e. empirical equations) for geostatistical or stochastic simulations are presented (lines 252-533).**

**Section 6.3 explores the significance of the estimated Transimissivity-Specific Capacity relationships (equations 7 and 8). Then criteria and precautions are suggested for the adoption with both equations in the hydraulic parametrization of alluvial aquifers. Unfortunately, such type of relationships are still required at many places because of the incomplete investigation techniques adopted in the past for aquifer parametrization.**

**Section 6.4 explores the obtained groundwater modeling results. This section describes what numerical model results tell us about aquifer processes and the implications on both groundwater level and quality. We will delete the most general statements from the discussion following the suggestions by the reviewer. Differences and similarities, with respect to previous published groundwater models, are stressed as well.**

**Section 6.5 explores the results of the scenarios simulations. This is significant because hydrogeological studies generally show very site-specific results.**

RC: Section 4.4 (Model Calibration) is weak. No relevant comments about the methodology used for calibration.

**AC: Section 4.4 clearly states how model calibration is achieved (i.e. by means of inverse procedure and by using observation points). It is largely known that the calibration of a groundwater flow model is performed by varying the values of one or more parameters (i.e. hydraulic conductivity, storativity) until simulation results and measured values agree. The results of the calibration in terms of quality of the results is reported (lines 418-422 and 449-455 for steady and transient model, respectively).**

RC: No relevant comments are focused to justify the obtained model calibration statistics.

**AC: We will add some comments about obtained calibration values.**

RC: No comparison between the initial (conceptual model) and calibrated parameters (K, Ss, Recharge, etc.). No comparison between initial groundwater budget derived from conceptual model statement (in the whole or in a certain period) respect the obtained by modeling. Some of these issues are weakly illustrated in some parts of the text and in some figures, but I consider that an integrative and detailed explanation is needed.

**AC: The comparison between initial and calibrated parameters is shown in Table 4. Recharge has not been calibrated, hence no comparison is required in the discussions section.**

No sensitivity analysis of the parameters used in the model is commented.

**We agree with the reviewer. No sensitivity analysis is presented. It could be interesting to stress the sensitivity of hydraulic parameters even if not a major scientific aim. We will add some sentences in the results section.**

RC: It is interesting to state any trends of hydrochemistry data variation along time, related to changes in land uses and anthropogenic factors. And a specific discussion of the effect of contamination/pollution/human factors to data is desired.

**AC: This is not the aim of the paper. For details about hydrogeochemical characterization of the study area see https://doi.org/10.1016/j.jhydrol.2017.02.025**

**Some sentences about the possible effects of groundwater rebound on groundwater quality in urban settings are presented in the introduction and re-discussed in the discussions. Actually, this is part of a forthcoming work.**

RC: Authors have to explain how the reconstruction of aquifer geometry can be assessed by means of groundwater modeling and justified by means of a sensitivity analysis.

**AC: See comments above. In any case the need for a robust hydrostratigraphic reconstruction cannot be by-passed by simply looking at model sensitivity.**

RC: Conclusions are too general and obvious. The authors do not need to mention the results of the numerical model in the conclusion part.

**AC: The conclusions section describes the progress with respect to the available research and the critical elements of the proposed investigation, including the most relevant results. This is what commonly accepted in the writing of scientific papers. Accordingly, the comparison between historical groundwater level pattern and future scenarios (i.e. lines 625-629) in our opinion can be considered a critical element.**

RC: Authors could use the last section of the paper to extend their discussion to make clear the new contributions that the manuscript supports. Include important implications of your work on the scientific community and also for local groundwater management.

**AC: We agree with the reviewer. We will modify the conclusions section by including some general implications for the scientific community and for local groundwater management. In any case, we consider the discussions different from the conlcusions.**

---

## Referee Comment (RC2) · Anonymous Referee #2 · 13 Feb 2018

This paper presents a comprehensive approach to implement the hydrostratigraphic reconstruction of multi-aquifers and corresponding climate change dependent groundwater flow modeling in Milan Po Plain area of Northern Italy. In my opinion, the topics of this paper might be of interest to the readers of this journal, but it cannot be considered acceptable for publication in its current state.

1. A major concern is that the paper does not appear to be significantly innovative, but consists of a complex exercise that applies varieties of approaches well established in the literature. The novelty of this paper should be reinforced to illustrate the scientific

and academic findings in the study.

2. Another important concern relates to the groundwater flow modeling. Although the simulation model is generally well calibrated and technically sound, the uncertainty associated with recharge should be considered in simulating future scenarios. As shown in Section 4.2, recharge for different subdomains was obtained by different methods/models and has not been calibrated in the groundwater flow model. Actually, groundwater flow and its level should be mostly dominated by recharge. Then the recharge should be evaluated to proof the value of the modeling work for improving the reliability of groundwater level under changing climatic conditions. To some extent, the change of direct recharge may be more sensitive to the modeling results than the value of indirect precipitation due to climate change described in the paper. So, I strongly recommend authors investigate the sensitivity analysis regarding the fitting parameters of the model and the input values to the model.

---

## Author Comment (AC2) · 9 Mar 2018

This paper presents a comprehensive approach to implement the hydrostratigraphic reconstruction of multi-aquifers and corresponding climate change dependent groundwater flow modeling in Milan Po Plain area of Northern Italy. In my opinion, the topics of this paper might be of interest to the readers of this journal, but it cannot be considered acceptable for publication in its current state.

1. A major concern is that the paper does not appear to be significantly innovative, but consists of a complex exercise that applies varieties of approaches well established in the literature. The novelty of this paper should be reinforced to illustrate the scientific and academic findings in the study.

**AC: We believe that the paper novelty does not rely in a single innovative idea, but on the formalization and testing of a rigorous hydrostratigraphic approach for the characterisation and modelling of aquifers in mixed urban and agricultural area. This approach is demonstrated for a complex case study and may be followed and replicated by other researchers in this and in similar areas. Thus, we believe that the paper may have significant scientific and practical impacts.**

**Here, we will discuss the main strengths of the paper and their novelty.**

**(1) The adoption of a hierarchical classification of hydrological units is not the first attempt in the literature. In this paper, we demonstrate how a sedimentological analysis of the stratigraphic data is required to avoid simplified and unreliable approaches and to help grouping lithological layers in significant hydrostratigraphic units (. In fact, the attribution of hydraulic parameters following the qualitative description of single layers as from the drilling phases is often affected by the subjectivity in the description. Recognition of specific sedimentological and stratigraphic sequences adds meaning and robustness to this work.**

**(2) For the hydraulic parametrization, new Transmissivity-Specific Capacity relationships (equations 7 and 8) are presented and compared to those proposed in the literature. Such relationships may be adopted for similar aquifers where such type of relationships are still required because of the incomplete investigation techniques adopted in the past for aquifer parametrization. This empirical approach, which uses regression analysis of specific capacity and transmissivity values from pumping tests from the same aquifer is a viable alternative to the analytical approach. However, a limitation of the method for estimating transmissivity from specific capacity is that drawdown in a well is function of well efficiency and losses, which for high pumping rates may be substantial. In fact, this method generally tends to under-predict transmissivity (Razack and Huntley, 1991). Thus, the error needs to be considered in the context of few orders of magnitude of transmissivity. For this reason, the parametric bootstrapping technique has been used to generate bootstrap samples of T-Sc, both from well and step-drawdown tests data to estimate uncertainty of the proposed equations. In particular, the results (i.e. $A_1$ and D parameter of equation 5) obtained for the step-drawdown tests are affected by larger uncertainties (Fig. 8) and larger confidence intervals (10th-90th percentiles) with respect to the ones obtained for the well tests. As demonstrated by Meier et al. (1999), this larger uncertainty can be related on the analysed time test duration. In fact, slopes and regression coefficients are smaller and the intercept is larger for late time than for early time data. This is also visible in Fig. 8a, where the $T-S_c$ data points for step-drawdown tests (Fig. 8) show a higher degree of scatter with respect to $T-S_c$**

**data points for well-tests. In any case, the proposed empirical relations are similar to those available for heterogeneous sandy and gravelly sediments.**

**(3) Finally, the investigation of future climate scenarios allows to quantify the potential effects of global change on the groundwater levels and the water budget. As shown by several studies, this effect is very site-specific and climate-model-specific. In our paper, we demonstrate that, for the case study of Milano, the impact of climatic change is secondary with respect to anthropogenic stresses, at least for short/medium-time scenarios (20 years), and that it should not affect water resources in the near future. This is an important and novel finding that has significant practical effects for a densely populated area such as Milan. At the same time, we suggest that more complex interactions can arise from combinations of scenarios (e.g. groundwater use for irrigation, change in crop type).**

**In the revised version of the paper, we modified the introduction and discussion to strengthen scientific and academic findings and to keep the focus on discussing the novel elements, as discussed above.**

 2. Another important concern relates to the groundwater flow modeling. Although the simulation model is generally well calibrated and technically sound, the uncertainty associated with recharge should be considered in simulating future scenarios. As shown in Section 4.2, recharge for different subdomains was obtained by different methods/models and has not been calibrated in the groundwater flow model. Actually, groundwater flow and its level should be mostly dominated by recharge. Then the recharge should be evaluated to proof the value of the modeling work for improving the reliability of groundwater level under changing climatic conditions. To some extent, the change of direct recharge may be more sensitive to the modeling results than the value of indirect precipitation due to climate change described in the paper. So, I strongly recommend authors investigate the sensitivity analysis regarding the fitting parameters of the model and the input values to the model.

**AC: We agree with the reviewer. Also following the comments of the first reviewer, we performed a sensitivity analysis for the hydraulic conductivity and input variables (i.e. rainfall and recharge) to establish the effect of uncertainty on the calibrated groundwater flow model, and we include this analysis in the revised version of the manuscript. Here, we briefly anticipate method and results.**

**The sensitivity analysis was carried out on the calibrated model. Accordingly, the horizontal and the vertical hydraulic conductivity, and the recharge (i.e. rainfall infiltration and irrigation) values of each subunit were modified by specific percentage for the input variables (±25%, ±50%, and ±100% of the initial values) and by a multiplicative factor for the hydraulic conductivity (0.1, 0.2, 0.5, 2, 5, and 10, thus allowing to test three orders of magnitude of K). The model outputs of the different parameters sets (i.e. hydraulic heads at observation points) were compared to those resulting from calibrated model to understand the sensitivity of the model to such parameters (Figs. 1, 2).**

**This analysis is introduced in the revised manuscript in section 4.4 and in the related result/discussion sections. As shown in figures 1, the modelled groundwater system is mostly sensitive to groundwater abstractions and rainfall recharge. Another important result is that**

the model is not strongly sensitive to irrigation recharge, which is the most uncertain input in the study area.

The results of the sensitivity analysis on input parameters (i.e. recharge, irrigation, and withdrawals) can be analysed from a climate change point of view. For example, an increase of groundwater recharge may result both from "wet" and "cold" future scenarios (Allen et al., 2003) due to additional rainfall or a reduction of evapotranspiration triggering an increase of recharge, respectively. These scenarios, may lead to a further increase of groundwater level in the unconfined aquifer of about 2 m (considering an increase of 25%).

On the other hand, a dry scenario may lead to a decrease of recharge amount because most of the rainfall water is evaporated. Considering only changes in recharge rates, this may lead to a decrease of shallow groundwater levels (c.a. -2 m for a 25% reduction). Obviously, the irrigation requirements might need to be reconsidered to counteract the effect of dryness. This could lead to more groundwater withdrawals, which could lower the groundwater levels (c.a. -3 m up to -6 m, for a 25% increase), or to an increase of irrigation water volume (considering the fully gravity-driven irrigation system) which, in its turn could lead to a groundwater level increase (c.a. +3 m up to 6 m, for a 25% increase). Furthermore, according to modeling results a change in river discharge as a consequence of climate change can not be excluded (e.g. higher for rainy year and lower for droughty years such as for 2003 and 2014, respectively).

Overall, the sensitivity of the groundwater system depends on how precipitation, temperature, water demand for crop production, and withdrawals are related. Their combined effects might be cancelled out leading to moderate changes of the water balance components (*Woldeamlak et al., 2006*). This support the idea that both climatic and socio-economic changes are fundamental for simulating future scenarios. This is particularly significant because, even if the impact of climate on simulated recharge is expected to be more important, the socio-economic factors may produce highly significant regional changes especially in densely populated areas and where major changes to the temporal distribution of groundwater abstraction may occur (*Holman, 2006*; *Holman et al., 2011*). For this reason, we considered well established scenarios (i.e. *IPCC* climate projections, decrease of irrigation, and decrease of groundwater withdrawals according to demographic scenarios) for simulating medium-term scenarios (Section 4.5). For the adopted scenarios, in terms of duration, the changes in the recharge in the upper part of the catchments are not so relevant. In fact, isotopic data suggest a relative young groundwater age (i.e. between 20-30 years, *Gorla et al., 2016*).

Regarding the hydraulic conductivity, we observe that the groundwater model is particularly sensitive to changes in hydraulic conductivity of unconfined aquifer sub-units. Among these, proximal sectors of alluvial fans/megafans are the most sensitive zones. For these sectors, we also observe that extremely low values (i.e. 1 order of magnitude lower) lead to numerical instabilities. On the other side, the hydraulic heads are quite insensitive to changes in hydraulic conductivity values of the semi-confined aquifer and of the aquitard between unconfined and semi-confined aquifers. This is particularly relevant and it supports the idea that the aquitard does not affect much the behaviour of the hydrogeological system. The behaviour of the proximal unconfined aquifer (where most of the fontanili are located) is

apparently anomalous. In fact, a reduction of hydraulic conductivity results in a reduction of the hydraulic head in the whole model.

This may be the consequence of the imposed boundary conditions (i.e. Dirichlet condition based on hydraulic head surveys) and of an unreliable increase of discharge through the fontanili springs, which have been simulated as flux-constrained Dirichlet condition.

[Figure]

**Fig. 1 – Sensitivity of groundwater levels to model inputs: (a) Rainfall recharge, (b) Irrigation recharge, and (c) groundwater withdrawals. Results for the Milan city area has been distinguished (i.e. solid squares). The scatter plots summarize the differences (m) between calibrated hydraulic heads (i.e. at observation points) and hydraulic heads of sensitivity scenarios. Inputs variability fields, according to simulated climate, irrigation, and pumping scenarios (section 4.5) are reported (red lines).**

[Figure]

**Fig. 2 – Box plots showing the sensitivity of groundwater levels to hydraulic conductivity for (a) proximal sectors (i.e. proximal fringes of fan deposits) of unconfined aquifer, (b) distal sectors of unconfined aquifer (i.e. distal fringes of fan deposits), (c) semi-confined aquifers, and (d) aquitard between semi-confined and unconfined aquifers.**